# Longitudinal EEG power in the first postnatal year differentiates autism outcomes

Laurel J. Gabard-Durnam[1], Carol Wilkinson[1], Kush Kapur[2], Helen Tager-Flusberg[3], April R. Levin[1,2] & Charles A. Nelson[1,4,5]

An aim of autism spectrum disorder (ASD) research is to identify early biomarkers that inform ASD pathophysiology and expedite detection. Brain oscillations captured in electroencephalography (EEG) are thought to be disrupted as core ASD pathophysiology. We leverage longitudinal EEG power measurements from 3 to 36 months of age in infants at low- and high-risk for ASD to test how and when power distinguishes ASD risk and diagnosis by age 3-years. Power trajectories across the first year, second year, or first three years postnatally were submitted to data-driven modeling to differentiate ASD outcomes. Power dynamics during the first postnatal year best differentiate ASD diagnoses. Delta and gamma frequency power trajectories consistently distinguish infants with ASD diagnoses from others. There is also a developmental shift across timescales towards including higher-frequency power to differentiate outcomes. These findings reveal the importance of developmental timing and trajectory in understanding pathophysiology and classifying ASD outcomes.

---

[1] Division of Developmental Medicine, Boston Children's Hospital, Harvard Medical School, 1 Autumn Street, Boston, MA 02215, USA. [2] Department of Neurology, Boston Children's Hospital, Harvard Medical School, 300 Longwood Avenue, Boston, MA 02215, USA. [3] Department of Psychological and Brain Sciences, Boston University, 64 Cummington Mall, Boston, MA 02215, USA. [4] Department of Pediatrics, Harvard Medical School, 300 Longwood Avenue, Boston, MA 02215, USA. [5] Harvard Graduate School of Education, 13 Appian Way, Cambridge, MA 02138, USA. Correspondence and requests for materials should be addressed to C.A.N. (email: Charles.nelson@childrens.harvard.edu)

Autism spectrum disorder (ASD) is a prevalent neurodevelopmental disorder that currently affects 1 out of 40 children[1]. Early identification and intervention is critical to improve outcomes for those with ASD, but overt behavioral symptoms of ASD generally only begin to manifest across the second year after birth[2–5]. Thus, there is a need for more sensitive physiological indicators of ASD that can be identified earlier in development[5,6]. As brain pathophysiology gives rise to the behavioral symptoms of ASD, measuring neurophysiology across the first years after birth facilitates detecting early biomarkers of subsequent ASD diagnosis that can expedite detection and treatment, and provide biological targets for novel interventions[7,8].

An emerging literature suggests that early disruptions in the brain's oscillatory rhythms are core neural features of ASD pathophysiology. These rhythms are captured by electroencephalographic (EEG) activity. Accordingly, many differences have been noted between groups at elevated risk for ASD or with ASD outcomes across a variety of EEG-derived oscillatory measures, including spectral power[9–12]. In particular, differences between ASD risk groups in EEG spectral power across a range of frequency bands over the frontal cortex have been shown to emerge early in infancy and relate to later symptomatology[9–16]. Infants with elevated risk of ASD have been shown to have lower frontal alpha and beta band power at 3 months of age, lower frontal power from delta through gamma frequency bands by 6 months of age, and different age-related changes in delta, alpha, beta, and gamma band power through two years of age, showing largely steeper age-related increases in power[9–11]. Compared to other cortical regions, frontal EEG power consistently demonstrates differences in theta, alpha, and gamma bands during the second year postnatally that have been associated with a range of ASD symptom domains in this elevated-risk population, including sensory hyporesponsiveness, cognitive deficits, language, and the degree of restricted and repetitive behaviors[9,10,12–16]. However, it remains to be determined how well early EEG differences can distinguish subsequent ASD outcomes.

Importantly, longitudinal studies have found significant developmental changes in frontal EEG power in all frequency bands across the first years after birth[9,10,13]. Thus, the capacity for EEG power to differentiate ASD outcomes and the integrity of individual EEG power measures may change across this period[8]. These critical factors have yet to be tested. Moreover, developmental changes in EEG power may themselves constitute diagnostic measures (e.g., [17–19]), but no study to date has tested whether EEG trajectories over development may be leveraged for differentiating ASD outcomes. Longitudinal comparisons are needed to determine when EEG power is most useful for differentiating ASD outcomes, and how candidate EEG power biomarkers change across early neurodevelopment.

The present study seeks to determine the EEG power pathophysiology over the first three postnatal years that differentiates subsequent ASD risk and diagnostic outcomes. High-risk populations that have elevated ASD incidence, such as infant siblings of children with ASD, facilitate prospective testing for early pathophysiology in ASD[4,5]. Here, we leverage longitudinal baseline EEG in a cohort of these high familial risk infant siblings and infants at low familial risk for ASD to derive data-driven profiles of EEG power across multiple timescales. Given the robust early differences observed over frontal cortex, the present study focuses a priori on frontal EEG power to address this question. We focus on frontal EEG pathophysiology across three key developmental windows: the first year, antecedent to behavioral symptoms; the second year, concurrent with emerging behavioral symptoms; and the three year period including the age of confirmed ASD diagnosis. We first assess the performance of EEG power measures across developmental windows to determine which period of time best differentiates ASD risk and diagnostic outcomes within our sample. We also perform supplemental analyses comparing frontal cortex EEG to whole-head coverage and temporal-parietal spatial configurations to test for spatial specificity in differentiation effects. Finally, we identify which EEG frequency bands and their developmental trajectories differentiate groups as potential diagnostic biomarkers, and whether these measures' identities change across the three developmental windows. In these ways, we aim to illuminate the EEG power measures and developmental timing that provide the most robust differentiation of ASD risk and diagnoses.

## Results

**Data-driven models of EEG power differentiate ASD outcomes.** To determine when across the first three years after birth EEG power measures best differentiate ASD risk and outcome groups within this sample, a series of logistic regression models were constructed. These logistic regression models differentiated groups (ASD vs. HRA−, ASD vs. LRC, and HRA− vs. LRC) using longitudinal EEG parameters from three developmental windows: 3–12 months, 12–24 months, and 3–36 months (Fig. 1, Table 1)). For each group comparison in each developmental window, the EEG power intercepts and developmental slope variables for each frequency band were potential model parameters. Data-driven model construction selected the set of EEG power parameters that best differentiated each pair of groups over each developmental window. Parental education and participant sex were included as covariates for each analysis. First, model performance was compared across the different developmental windows to identify the time period that best differentiated risk and outcome groups. The models constructed with frontal EEG were also compared to models using different spatial layouts (whole-head coverage and temporal-parietal channels; Table 1, Supplementary Tables 1–6). Finally, the selected EEG power parameters and their significance within each model were compared to identify candidate longitudinal power biomarkers across the different developmental windows.

We first compared how well the different data-driven models performed at differentiating groups across developmental windows by assessing receiver operating characteristic (ROC) curves for the models. Notably, all models tested significantly differentiated each pair of groups (all area under the curve (AUC) bootstrapped confidence interval (CI$_{95}$) lower bounds >0.5, Fig. 1, Table 1). Comparisons of model performance are organized below by those discriminating children with ASD diagnoses from others (ASD vs. HRA−, ASD vs. LRC) across timescales, and those discriminating children with differential ASD risk in the absence of ASD (HRA− vs. LRC) across timescales.

**ASD diagnostic outcome discrimination across development.** Across developmental windows and spatial topographies, the frontal 3–12 month EEG models most accurately discriminated the ASD group from the other groups (Table 1). These 3–12 month models also had higher specificity rates and rates predicting true positive ASD outcomes (positive predictive values (PPV)) than the other frontal models, as well as higher or comparable rates predicting true negative ASD outcomes (e.g., for the model distinguishing ASD from HRA− infants, frontal 3–12 month negative predictive value of 91.67 vs. 92.98 for the 3–36 month model; Table 1). No frontal EEG model consistently outperformed others with respect to sensitivity rates in the primary analyses, and all models achieved high sensitivity rates. As more infants contributed data to the frontal 3–36 month models than the frontal 3–12 month models, secondary analyses modeled a restricted 3-year subsample of the same infants from the 3–12 month analysis. However, the frontal 3–12 month models

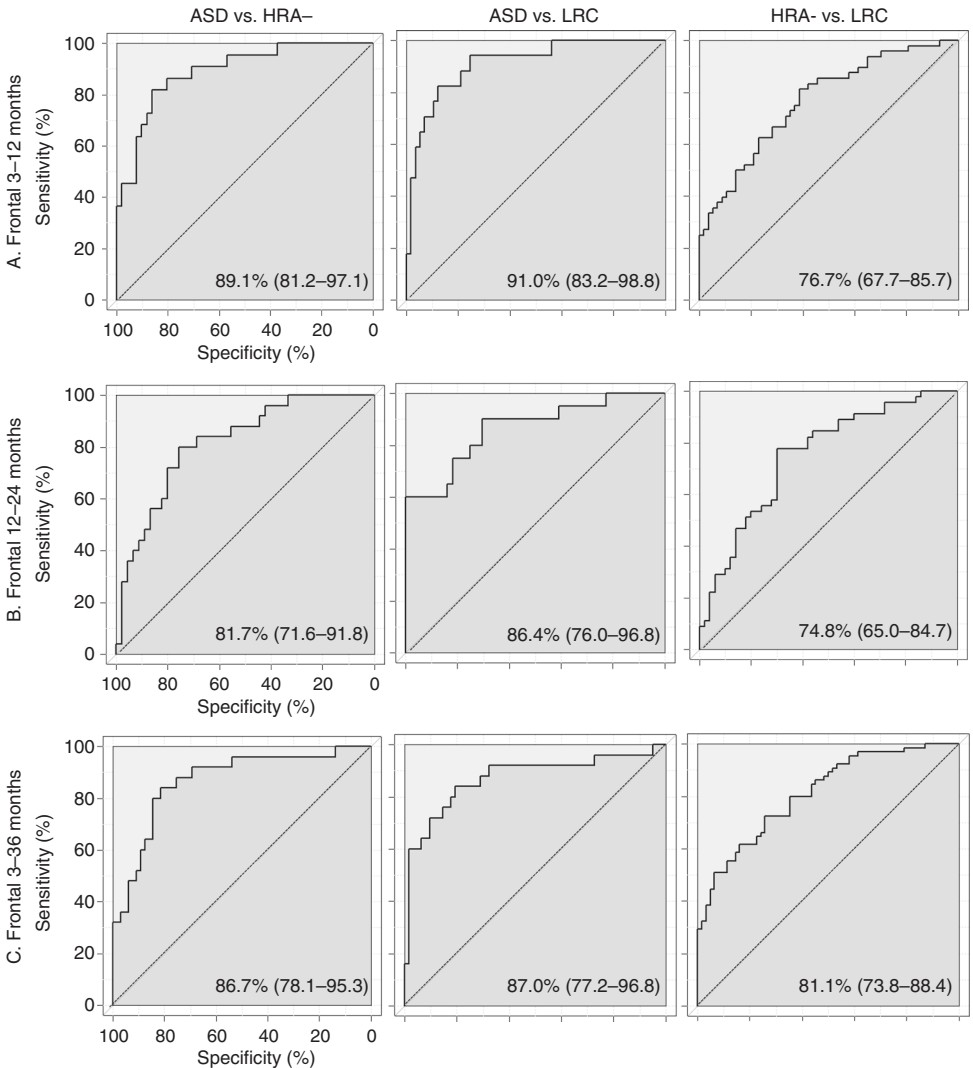

**Fig. 1** ROC curves. Receiver operating characteristic (ROC) curves for each frontal EEG model comparing pairs of ASD outcome groups for each developmental window (**a–d**). Area under the curve and their 95% confidence intervals are given in the right corner of each ROC curve. Dashed black lines indicate chance performance. Solid black line indicates each model's performance

remained the most accurate at differentiating those with ASD diagnoses from others.

**ASD-risk discrimination across development.** Across developmental windows and spatial topographies, the frontal 3–36 month EEG model most accurately discriminated the high-risk from low-risk groups in the absence of ASD (Table 1). The frontal 3–36 month model also had higher optimized specificity rates and rates predicting true high-risk status, but lower sensitivity rates and rates predicting true low-risk status than the other windows of frontal EEG power (Table 1). The risk-discrimination models achieved poorer discrimination accuracy, sensitivity, specificity, and true negative prediction rates than the corresponding ASD outcome discrimination models over all developmental windows.

**Identification of EEG power candidate biomarkers.** Next, to assess which frontal EEG power measures differentiated these ASD groups across developmental timescales as potential biomarkers, the EEG power parameters that the data-driven model construction process had selected were examined for each group comparison across each developmental window (Tables 2–4;

whole-head and temporal-parietal model parameters provided in Supplementary Tables 1–6). EEG power for a given frequency band in these models reflects the summed power across all frequencies within that frequency band (summed power). Parameters within each model were also evaluated to identify significant individual outcome discriminators. With regard to the participant sex and parental education covariates modeled, participant sex was not a significant measure in any model at any age. The parental education covariate was included in many models differentiating the LRC infants from others, but in the majority of comparisons, only the parameter differentiating parents with 4-year college degrees from those with further education was significant (education parameter 2). Supplemental analyses without the sex and education covariates returned similar results to the analyses that included these covariates. Comparisons for each developmental window are organized by those discriminating children with ASD diagnoses from others (ASD vs. HRA−, ASD vs. LRC) and those discriminating children in both high-risk groups relative to the low-risk children (i.e., high-risk markers, ASD vs. LRC and HRA− vs. LRC). All significance tests of individual model parameters refer to Student's *t* tests performed within the multiple logistic regression model (i.e., including all other model parameters simultaneously).

**Table 1 Model performance metrics discriminating autism groups**

| Models | Discrimination rate (CI$_{95}$) | Sensitivity[†] | Specificity[†] | PPV | NPV |
|---|---|---|---|---|---|
| **3–12 months** | | | | | |
| *Frontal* | | | | | |
| ASD vs. HRA- | 89.1% (81.2–97.1%) | 81.82 | 86.27 | 72 | 91.67 |
| ASD vs. LRC | 91.0% (83.2–98.8%) | 82.35 | 87.72 | 66.67 | 94.34 |
| HRA- vs. LRC | 76.7% (67.7–85.7%) | 81.25 | 61.4 | 63.93 | 79.55 |
| *Whole-head* | | | | | |
| ASD vs. HRA− | 81.7% (70.0–93.5%) | 90.91 | 64.71 | 52.63 | 94.29 |
| ASD vs. LRC | 80.3% (65.9–94.7%) | 76.47 | 80.7 | 54.17 | 92 |
| HRA− vs. LRC | 73.9% (64.6–83.3%) | 54.17 | 82.46 | 72.22 | 68.12 |
| *Temporal-Parietal* | | | | | |
| ASD vs. HRA− | 83.4% (72.9–93.9%) | 72.73 | 86.27 | 69.57 | 88 |
| ASD vs. LRC | 82.2% (68.7–95.8%) | 64.71 | 96.49 | 84.62 | 90.16 |
| HRA- vs. LRC | 77.4% (68.6–86.3%) | 64.58 | 78.95 | 72.09 | 72.58 |
| **12–24 months** | | | | | |
| *Frontal* | | | | | |
| ASD vs. HRA− | 81.7% (71.6–91.8%) | 80 | 75.56 | 64.52 | 87.18 |
| ASD vs. LRC | 86.4% (76.0–96.8%) | 90 | 70.45 | 58.06 | 93.94 |
| HRA− vs. LRC | 74.8% (65.0–84.7%) | 77.78 | 70 | 70 | 77.78 |
| *Whole-head* | | | | | |
| ASD vs. HRA- | 87.6% (78.8–96.5%) | 70 | 93.48 | 82.35 | 87.76 |
| ASD vs. LRC | 89.0% (78.9–99.1%) | 85 | 87.5 | 73.91 | 93.33 |
| HRA- vs. LRC | 80.0% (71.2–88.8%) | 58.70 | 87.5 | 81.82 | 68.85 |
| *Temporal-Parietal* | | | | | |
| ASD vs. HRA− | 76.9% (66.2–87.6%) | 80 | 65.31 | 54.05 | 86.49 |
| ASD vs. LRC | 89.4% (81.3–97.4%) | 90 | 75 | 60 | 94.74 |
| HRA- vs. LRC | 80.9% (72.6–89.2%) | 83.67 | 68.52 | 70.69 | 82.22 |
| **3–36 months** | | | | | |
| *Frontal* | | | | | |
| ASD vs. HRA− | 86.7% (78.1–95.3%) | 84 | 81.54 | 63.64 | 92.98 |
| ASD vs. LRC | 87.0% (77.2–96.8%) | 84 | 80.65 | 63.64 | 92.59 |
| HRA− vs. LRC | 81.1% (73.8–88.4%) | 72.31 | 74.19 | 74.6 | 71.88 |
| *Whole-head* | | | | | |
| ASD vs. HRA− | 87.7% (79.5–95.9%) | 84 | 84.62 | 67.74 | 93.22 |
| ASD vs. LRC | 85.7% (74.9–96.6%) | 80 | 90.32 | 76.92 | 91.80 |
| HRA− vs. LRC | 77.5% (69.4–85.5%) | 69.23 | 75.80 | 75 | 70.15 |
| Temporal-Parietal | | | | | |
| ASD vs. HRA− | 83.6% (75.1–92.2%) | 67.74 | 85.92 | 67.74 | 85.92 |
| ASD vs. LRC | 87.0% (77.5–96.6%) | 84 | 83.87 | 67.74 | 92.86 |
| HRA− vs. LRC | 75.2% (66.8–83.7%) | 72.31 | 72.58 | 73.44 | 71.43 |

[†]Evaluated at Youden's index maximum
*CI$_{95}$* 95% confidence interval around the estimate, *PPV* positive predictive value (true positive outcome percentage), *NPV* negative predictive value (true negative outcome percentage)

**Frontal EEG candidate biomarkers from 3–12 months.** Across models in this window, low frequency bands were significant individual parameters differentiating ASD outcomes (Table 2). Higher frequency bands, beta and gamma, were only selected in data-driven models differentiating ASD infants from others, such that lower levels of high-frequency power at age 6 months increased the log-odds of the child having ASD over and above the other selected frequency band parameters (e.g. beta power intercept, Student's $t$ test $p = 0.022$). Delta and theta slopes during this period were significant individual indicators of high-risk for ASD (ASD and HRA−) relative to LRC status, with steeper delta and less-steep theta slopes over the first postnatal year increasing the log-odds of the child belonging to the high-risk groups in the context of the other selected frequency band parameters (ASD: delta slope Student's $t$ test $p = 0.006$, theta interaction Student's $t$ test $p = 0.013$; HRA−: delta slope Student's t-test $p = 0.022$, theta slope Student's $t$ test $p = 0.003$).

**Frontal EEG candidate biomarkers from 12–24 months.** Across second-year models, significant EEG parameters occurred at higher frequencies than in the 3–12 month models (Table 3). Beta

frequency predictors were only selected in models differentiating the ASD children from others. Lower summed power in beta band at 12 months of age increased the log-odds of the child having ASD in the context of the other selected frequency band parameters (Student's $t$ test $p = 0.016$). Steeper slopes in beta power over the second postnatal year as a function of beta power at 12 months of age also increased the log-odds of the child having ASD within the high-risk group over and above the other selected frequency band parameters (Student's $t$ test $p = 0.033$).

**Frontal EEG candidate biomarkers from 3 to 36 months.** Higher frequencies were significant individual parameters relative to those in earlier windows (Table 4). Low alpha and gamma developmental slopes across 3 years were unique significant indicators of ASD diagnosis. Less steep increases in low alpha summed power from 3 to 36 months of age increased the log-odds of the child having ASD within the high-risk group, and the summed power in low alpha at 6 months of age interacted differently with the increase in low alpha power during this window to increase the log-odds of the child having ASD in the context of the other selected frequency band parameters (vs. HRA−: slope Student's

**Table 2 Frontal 3–12 months EEG power models**

| Parameters | ASD vs. HRA− B coefficient (SE) p value | | ASD vs. LRC B coefficient (SE) p value | | HRA− vs. LRC B coefficient (SE) p value | |
|---|---|---|---|---|---|---|
| Model intercept | −10.44 (5.91) | 0.077 | 10.36 (6.64) | 0.118 | **8.98 (2.43)** | **0.0002** |
| Sex | 1.23 (0.81) | 0.129 | – | – | – | – |
| Parental Education 1 | – | – | −0.99 (1.46) | 0.498 | −1.76 (0.95) | 0.065 |
| Parental Education 2 | – | – | **−3.34 (1.49)** | **0.025** | **−2.22 (0.87)** | **0.01** |
| *6-month Intercept* | | | | | | |
| Delta | **−25.5 (12.59)** | **0.043** | 24.98 (15.62) | 0.11 | – | – |
| Theta | 20.4 (12.17) | 0.094 | −24.79 (15.01) | 0.099 | – | **–** |
| Low Alpha | **15.72 (7.71)** | **0.042** | – | – | **−5.85 (1.77)** | **0.001** |
| High Alpha | 9.93 (8.25) | 0.229 | **17.78 (7.28)** | **0.015** | – | – |
| Beta | −8.97 (5.75) | 0.119 | **−19.25 (8.38)** | **0.022** | – | – |
| Gamma | – | – | −2.91 (6.38) | 0.648 | – | – |
| *Slope 3–12 months* | | | | | | |
| Delta | – | – | **30.78 (11.11)** | **0.006** | **6.18 (2.69)** | **0.022** |
| Theta | **38 (18.07)** | **0.035** | −8.31 (10.68) | 0.436 | **−8.81 (3.02)** | **0.003** |
| Low Alpha | −26.02 (16.16) | 0.107 | – | **–** | – | – |
| High Alpha | 31.6 (18.45) | 0.087 | – | – | – | – |
| Beta | −29.73 (17.57) | 0.091 | – | – | – | **–** |
| Gamma | 5.66 (3.85) | 0.142 | −17.39 (9.71) | 0.073 | – | – |
| *Intercept × Slope* | | | | | | |
| Delta | – | **–** | – | **–** | – | – |
| Theta | −22.43 (12.71) | 0.078 | **−18.86 (7.57)** | **0.013** | – | – |
| Low Alpha | 21.74 (11.75) | 0.064 | – | – | – | – |
| High Alpha | **−35.59 (17.32)** | **0.04** | – | – | – | – |
| Beta | 18.39 (11.7) | 0.116 | – | – | – | **–** |
| Gamma | – | **–** | 13.62 (8.59) | 0.113 | – | – |

SE standard error; bold values indicate statistically significant parameters (determined by Student's *t* test) within each model at the level of *p* < 0.05

**Table 3 Frontal 12–24 months EEG power models**

| Parameters | ASD vs. HRA− B coefficient (SE) p value | | ASD vs. LRC B coefficient (SE) p value | | HRA- vs. LRC B coefficient (SE) p value | |
|---|---|---|---|---|---|---|
| Model intercept | **−14.98 (5.69)** | **0.008** | **21.12 (7.69)** | **0.006** | **11.94 (3.13)** | **0.0001** |
| Sex | – | – | 1.6 (0.84) | 0.056 | – | – |
| Parental Education 1 | – | – | −2.02 (1.55) | 0.193 | – | – |
| Parental Education 2 | – | – | **−3.21 (1.38)** | **0.02** | – | – |
| *12-month Intercept* | | | | | | |
| Delta | **22.49 (9.32)** | **0.016** | −10.34 (6.5) | 0.112 | – | – |
| Theta | **−23.93 (9.56)** | **0.012** | – | **–** | – | **–** |
| Low Alpha | **11.17 (5.59)** | **0.046** | – | – | – | – |
| High Alpha | – | – | 13.69 (7.77) | 0.078 | **−9.71 (2.54)** | **0.0001** |
| Beta | −0.19 (2.85) | 0.946 | **−14.19 (5.88)** | **0.016** | – | – |
| Gamma | – | – | – | – | – | – |
| *Slope 12–24 months* | | | | | | |
| Delta | −13.59 (8.68) | 0.117 | **−37.42 (16.06)** | **0.02** | – | – |
| Theta | – | **–** | – | **–** | – | – |
| Low Alpha | 6.15 (3.78) | 0.104 | – | **–** | – | – |
| High Alpha | **−10 (4.8)** | **0.037** | – | – | **−9.37 (3.66)** | **0.01** |
| Beta | **31.54 (11.98)** | **0.008** | – | – | – | **–** |
| Gamma | – | – | – | **–** | – | – |
| *Intercept × Slope* | | | | | | |
| Delta | 9.89 (6.2) | 0.111 | **26.59 (11.74)** | **0.024** | – | – |
| Theta | – | – | – | – | – | – |
| Low Alpha | – | – | – | – | – | – |
| High Alpha | – | **–** | – | – | 4.49 (3.17) | 0.156 |
| Beta | **−15.93 (7.46)** | **0.033** | – | – | – | **–** |
| Gamma | – | **–** | – | **–** | – | – |

SE standard error; bold values indicate statistically significant parameters (determined by Student's *t* test) within each model at the level of *p* < 0.05

*t* test *p* = 0.012, vs. LRC: interaction Student's *t* test *p* = 0.042). Less steep changes in gamma power from 3–36 months increased the log-odds of the child having ASD across comparisons, over and above the other selected frequency band parameters (vs. HRA−,

Student's *t* test *p* = 0.031; vs. LRC: Student's *t* test *p* = 0.028). A significant interaction between summed power in the beta band at 6 months and the developmental slope in summed power in the beta band over 3 years was a significant individual indicator of

**Table 4 Frontal 3–36 months EEG power models**

| Parameters | ASD vs. HRA− B coefficient (SE) p value | | ASD vs. LRC B coefficient (SE) p value | | HRA- vs. LRC B coefficient (SE) p value | |
|---|---|---|---|---|---|---|
| Model intercept | **−11.27 (4.25)** | **0.008** | 0.47 (7.39) | 0.95 | **11.03 (3.43)** | **0.001** |
| Sex | 1.05 (1.05) | 0.138 | – | – | – | – |
| Parental Education 1 | 1.29 (0.92) | 0.162 | −1.4 (1.27) | 1.27 | **−1.78 (0.86)** | **0.038** |
| Parental Education 2 | −0.62 (0.85) | 0.465 | **−2.72 (1.24)** | **0.028** | **−2.06 (0.76)** | **0.007** |
| 6-month Intercept | | | | | | |
| Delta | 8.17 (10.9) | 0.454 | 11.99 (13.91) | 0.389 | – | – |
| Theta | 9.44 (12.22) | 0.44 | −15.63 (15.82) | 0.323 | −1.83 (3.21) | 0.57 |
| Low Alpha | **−16.48 (6.75)** | **0.015** | 7.71 (8.9) | 0.386 | – | – |
| High Alpha | **16.46 (5.77)** | **0.004** | 2.16 (11.64) | 0.852 | −5.36 (4.35) | 0.218 |
| Beta | – | – | 0.68 (8.45) | 0.936 | −1.44 (3.01) | 0.633 |
| Gamma | **−7.34 (3.34)** | **0.028** | −7.77 (5.88) | 0.186 | – | – |
| Slope 3–36 months | | | | | | |
| Delta | **74.03 (74.03)** | **0.031** | 83.66 (50.61) | 0.098 | 9.35 (5.64) | 0.097 |
| Theta | −39.15 (−39.15) | 0.221 | −121.48 (65.78) | 0.065 | 17.9 (13.17) | 0.174 |
| Low Alpha | **−20.6 (8.2)** | **0.012** | **74.56 (37.72)** | **0.048** | – | – |
| High Alpha | – | – | −59.94 (44.13) | 0.174 | −25.01 (13.67) | 0.067 |
| Beta | 7.4 (5.34) | 0.166 | **71.93 (29.12)** | **0.014** | **36.07 (14.95)** | **0.016** |
| Gamma | **−10.64 (4.92)** | **0.031** | **−24.7 (11.23)** | **0.028** | – | – |
| Intercept × Slope | | | | | | |
| Delta | **−56.39 (26.88)** | **0.036** | −54.61 (37.77) | 0.148 | – | – |
| Theta | **56.61 (27.9)** | **0.042** | 95.13 (52.8) | 0.072 | **−23.54 (9.97)** | **0.018** |
| Low Alpha | – | – | **−55.91 (27.5)** | **0.042** | – | – |
| High Alpha | – | – | 47.25 (36.4) | 0.194 | 25.74 (13.33) | 0.053 |
| Beta | – | – | **−46.04 (18.58)** | **0.013** | **−28.13 (11.16)** | **0.012** |
| Gamma | – | – | 15.94 (8.79) | 0.07 | – | – |

*SE* standard error; bold values indicate statistically significant parameters (determined by Student's *t* test) within each model at the level of *p* < 0.05

high-ASD risk relative to LRC status (ASD: Student's *t* test *p* = 0.013, HRA−: Student's *t* test *p* = 0.012).

## Discussion

One major unresolved goal in ASD research is the identification of early biomarkers that reveal the underlying pathophysiology differentiating subsequent diagnostic outcomes[8]. To address this challenge, we leveraged longitudinal baseline EEG across the first 3 years post-birth from the largest electrophysiological study of infants at low- and high-familial risk for ASD to date. We identified EEG power parameters across key developmental windows that distinguished risk and subsequent diagnostic outcomes with high fidelity. These findings provide evidence that EEG power measures constitute highly informative candidate biomarkers in the following ways.

This is the first study of potential neurophysiological biomarkers in ASD to assess diagnostic differentiation across ages within the same infants. Our findings comparing EEG power measures taken across multiple developmental windows and spatial configurations indicate that frontal EEG power within the first year after birth best discriminates ASD outcomes. EEG power closer to the age of diagnosis (when behavioral symptoms are evident) did not provide additional utility for differentiating outcomes, and instead showed diminished specificity and capacity to detect true ASD diagnoses at optimized thresholds. That is, the capacity of the EEG power measures studied here to classify ASD as a homogenous outcome appears to wane as the capacity of behavioral measures to do so grows[20]. Notably, although the prevalence of ASD differs between males and females, sex was not a significant predictor in any of the frontal EEG models, suggesting the information provided by the EEG parameters was more useful in differentiating ASD outcomes. Moreover, the results differentiating between the high- and low-ASD risk groups in the absence of ASD suggest that early frontal EEG power is sensitive enough to differentiate subclinical changes in brain function as well[21]. This capacity to differentiate between risk groups did improve over the 3 years of EEG measurements, suggesting subclinical differences may emerge more gradually or at a later time than differences in the high-risk infants with future ASD diagnoses.

Our findings also indicate that the spatial localization of EEG power measurements matters in distinguishing outcomes. The a priori clustered frontal EEG region of interest and a clustered temporal-parietal layout examined in supplemental analyses both provided better differentiation than the averaged 10–20 standard layout in the first postnatal year, suggesting sparse electrode configurations like those used in clinical settings currently may not provide the optimal layout for measuring early EEG biomarkers of ASD. The more densely clustered EEG layouts may have benefited from a higher signal-to-noise ratio than the whole-head layout. Notably, the frontal region of interest also provided better differentiation in the first year than the clustered temporal-parietal region, suggesting the spatial specificity of the EEG power measures is important for discriminating groups. At later developmental windows, the densely clustered and whole-head configurations offered different strengths, such that the frontal and temporal-parietal layouts largely achieved higher sensitivity rates, while the whole-head layout had largely higher specificity rates. Thus, both the spatial location and the spatial density of EEG channels are important factors to consider in EEG-derived biomarker development to differentiate ASD outcomes, especially early in postnatal development.

Our data-driven modeling approach to identify pathophysiology highlights the importance of characterizing the longitudinal development of candidate biomarkers in several ways. First, this approach selected the most parsimonious set of parameters that differentiated groups within each developmental window, allowing us to examine whether the most robust EEG power discriminators changed as a function of age. For example, differences in EEG power observed in the first year may persist but may no

longer be the most robust discriminators in later developmental windows as other developmental dynamics emerge, including adaptive and compensatory changes. Indeed, across timescales from 12 months to 3 years, we observed developmental changes in which parameters provided the best differentiation. Though early emerging differences largely remained robust discriminators at later ages, we also noted a general shift from significant low frequency predictors across the first year toward additional significant higher frequency predictors across the second and third years. This delay in high-frequency candidate biomarkers showing robust differentiation recapitulates maturation patterns of EEG power spectra observed in prior studies during this same period[22].

In addition, our findings show that both EEG power in early infancy and subsequent developmental changes in power over months and years are critical pathophysiological features distinguishing ASD outcomes. Although prior studies have largely focused on evaluating candidate biomarkers at single timepoints, the data-driven selection of slope parameters in our models demonstrates that developmental trajectories themselves are robust potential biomarkers. This result adds to emerging evidence across modalities that developmental trajectories may be more informative features for distinguishing ASD outcomes than single time-point measures[17,18,23]. These EEG power developmental slope parameters also present robust biological opportunities for future interventions to target and alter the developmental course of ASD.

Importantly, we identified the EEG power pathophysiology during each developmental window that best differentiates ASD risk and subsequent ASD outcomes in our sample. As we tested for the combination of EEG power variables that together could distinguish groups, the set of power parameters included in each data-driven model constitutes the candidate biomarker for that developmental window. Almost all models included power parameters from multiple frequency bands, suggesting the pattern of power across bands facilitates better differentiation of risk and outcome groups than parameters in any single frequency band. These results are consistent with Wang and colleagues' synthesis across studies of baseline EEG power suggesting that an atypical distribution of power across the spectrum distinguishes individuals with ASD from neurotypical individuals[24]. Accordingly, recent interventions seeking to modulate frontal EEG power in those with ASD have assessed patterns of change across the power spectrum as the biological target[25]. Evaluating multiple EEG frequency bands simultaneously may thus facilitate both efforts to differentiate outcomes and to leverage EEG power in interventions at the physiological level.

Across models, we also identified trends for specific EEG frequency bands that significantly contributed to group differentiation and may reflect physiological mechanisms altered in ASD development. We found that developmental changes in low-frequency power stratified both ASD risk status and diagnostic outcome. Specifically, we found that steeper rates of increase in summed power in the delta band across the first 12 months differentiated high-risk (regardless of diagnosis) from the low-risk status infants. Further changes in delta power over the next 2 years contributed to ASD outcome differentiation within the high-risk infants. Infants with subsequent ASD diagnoses had steeper rates of increase in summed delta power before age 3 years. Prior research has similarly noted low frequency power differences in children with ASD[26,27]. Activity in low frequency bands, including delta, has been shown to stimulate inhibition (i.e., activity suppression) in brain circuits[28]. Insufficient neural inhibition levels have been posited in ASD, and the increase in summed delta power observed in the high-risk infants may reflect an early compensatory mechanism to modulate excessive excitation levels and restore the balance of neural excitation and

inhibition[29,30]. Further research is needed to clarify the role that delta power modulation plays as a risk or resilience factor in ASD development.

We also found that across comparisons in the first three years, summed power in the gamma frequency band uniquely distinguished children with ASD relative to others. The high-risk infants who later received an ASD diagnosis demonstrated lower frontal gamma power at age 6 months and a lower rate of increase through age 3 years compared to the other groups. Prior work has identified similar gamma differences in ASD populations[31,32], and here for the first time we show that gamma changes differentiate diagnostic outcome. Notably, gamma power reflects the balance of excitation and inhibition within brain circuits that regulates neuroplasticity and experience-dependent development[30,33]. Lower gamma power in infants who are later diagnosed is consistent with insufficient neural inhibition and atypical neuroplasticity in ASD[34,35]. Moreover, gamma power has been associated with the development of language skills, an experience-dependent process that is often delayed and impaired in ASD[36,37]. Therefore, lower frontal summed gamma power during infancy in individuals with ASD may reflect core underlying neuroplasticity changes and portend emerging language impairments.

It is important to acknowledge several limitations of the current study within the context of biomarker development and validation. Here, we have identified candidate EEG power biomarkers across multiple developmental windows and spatial configurations, so further research is now required to evaluate the reliability, disorder specificity, and external validity of these measures. We chose to examine baseline EEG power measures as candidate biomarkers in part because baseline EEG power has previously demonstrated high test–retest reliability (Cronbach's alpha estimates of 0.90 and higher) across multiple time scales (within session, across days) in both typically developing children and clinical populations, including ASD (e.g., [38–42]). Prior studies have shown that 40–60 s of clean EEG data is sufficient to achieve such high levels of reliability in absolute power estimates across the scalp[39,40]. Therefore, in the present study, we required infants to contribute at least 40 s of clean EEG data for analyses to increase the reliability of our model estimates. Future research conducting test–retest analyses of EEG power measures within-participants across consecutive days in infancy is needed to confirm that the high reliability of these EEG power measures extends to the first postnatal months of age.

Second, further testing across clinical populations is required to determine the specificity of these EEG power candidate biomarkers. The present study could not disentangle whether the EEG power parameters differentiating the infants with ASD were specific to the familial-risk group or would generalize to other ASD-risk groups. In addition, the candidate biomarkers we identified may reflect disruptions in processes affected in other neurodevelopmental disorders as well (e.g., language acquisition[11]). However, infants with high-familial risk who are not diagnosed with ASD frequently manifest subclinical and clinical disruptions in the same domains as the high-risk infants with ASD, and develop other disorders at elevated rates[21,43–47]. The candidate biomarkers we identified distinguishing the high-risk infants with and without ASD achieved very high levels of accuracy, though, suggesting some specificity for ASD diagnosis in the present study. Additional research across ASD-risk and neurodevelopmental disorder-risk populations is needed to confirm the diagnostic specificity of these EEG power measures before they may be used as diagnostic biomarkers.

Lastly, further research is required to assess the external validity of the candidate biomarkers identified in the present analyses. While this study had a greater number of infant siblings with ASD diagnoses relative to prior reports, the present sample

is still limited, so these findings should be replicated with larger samples in future research. We were well powered to achieve the study's primary goal of evaluating model performance within our longitudinal sample across different developmental windows, but our sample size prevented us from validating model performance in out-of-sample sets (e.g., split-half training and testing sets).

In addition, the study's over-recruitment of high-risk infants is a strength, but the ratio of high-risk to low-risk infants is therefore not representative of the general population. So, the positive and negative predictive values (NPV) for models comparing these groups may be different in the general population. Future research testing the external validity of these power measures should also examine how robust the patterns identified here are across EEG acquisition equipment and parameters. To facilitate this type of future assessment, here we used a brief, naturalistic baseline paradigm, as collecting several minutes of EEG at baseline may prove more scalable than collecting event-related EEG, and we used standardized, open-source processing software that we developed for use across various EEG systems and files with different acquisition parameters.

This study sought to provide the pathophysiological course of EEG power that distinguishes ASD outcomes. Our longitudinal approach highlights the importance of developmental timing and the significant role of neurodevelopmental change in classifying disorder outcomes. Taken together, these findings demonstrate great promise for EEG power measures in the first year postnatally as candidate biomarkers in ASD.

## Methods

**Participants**. Participants in the present study were part of a prospective, longitudinal investigation across the first 3 postnatal years of infants at high- and low-familial risk for ASD. Institutional review board (IRB) (i.e., the ethics regulatory committee in the USA) approval was obtained from Boston University and Boston Children's Hospital (IRB # X06-08-0374) prior to the start of the study. Written, informed consent was obtained from a parent or guardian prior to each infant's participation in the study.

Infants were designated high-risk for autism (HRA) if they had at least one older sibling with a community diagnosis of ASD that could not be attributed to a known genetic disorder (e.g., Fragile X syndrome). Infants were designated low-risk for autism controls (LRC) if they had a typically developing older sibling and no first- or second-degree family members with ASD. All infants included in the study had a gestational age of at least 36 weeks, no known prenatal or postnatal medical or neurological complications, and no known genetic disorders (e.g., Fragile X Syndrome, Tuberous Sclerosis Complex).

Final ASD outcomes were determined for all infants using the Autism Diagnostic Observation Schedule (ADOS) in conjunction with clinical best estimate. For children meeting criteria on the ADOS, or coming within three points of cutoffs, a Licensed Clinical Psychologist reviewed scores and video recordings of concurrent and previous behavioral assessments, and using DSM-V criteria provided a best estimate clinical judgment: typically developing, ASD, or non-spectrum disorder (e.g., ADHD, anxiety, and language concerns). By 36 months of age we identified three groups of infants: HRA infants with ASD (ASD), HRA infants without ASD (HRA−), and LRC infants without ASD (LRC). Of the 102 HRA infants contributing data for this study, 4 children had final outcome judgments at 18 months (1 ASD and 3 HRA−), 15 children had final outcome judgments at 24 months (3 ASD and 12 HRA−), and 83 children had final outcome judgments at 36 months (27 ASD and 56 HRA−). The 3 HRA children with HRA- outcomes determined at 18 months of age all had ADOS calibrated severity scores of 1 (minimal to no evidence of ASD), and were not statistical outliers within their HRA- group on any EEG power measure, so they were retained in analyses. Across outcome visits, 31 HRA children met criteria for ASD (ASD outcome group) and 71 were classified as no ASD (HRA− outcome group). None of the 69 LRC infants contributing data met criteria for ASD at outcome visits. The demographic composition of each outcome group is presented in Table 5. The ASD group had significantly more male children than the HRA− group (Pearson $X^2$ (1, 102) = 4.64, $p = 0.031$). The LRC group had higher mean parental education level than both the ASD group (Pearson $X^2$ (2, 87) = 12.65, $p = 0.002$) and the HRA− group (Pearson $X^2$ (2, 127) = 6.62, $p = 0.036$). Accordingly, sex and parental education parameters were included as potential covariates during model selection to ensure the EEG parameters explained the variance in outcome status unrelated to these demographic parameters. Supplemental analyses were also performed without the sex and education parameters as potential covariates, and results were highly consistent with the reported results.

EEG data were collected when infants were 3, 6, 9, 12, 18, 24, and 36 months of age. Not all infants contributed adequate EEG data at all timepoints. The sample size of usable EEG for each outcome group at each timepoint, sample sizes for each of the primary analyses run across timepoints, and the mean number of EEG recordings that each participant within each outcome group contributed to primary analyses are provided in Table 5. Across outcome groups, comparable numbers of EEG recordings were contributed per participant. For primary analyses (3–12, 12–24, and 3–36 months) the mean number of EEG recordings contributed per participant did not significantly differ between pairs of outcome groups in 8 of the 9 comparisons (only LRC and HRA− groups in the 3–36 month analysis significantly differed, with LRC infants contributing more EEGs [Student's $t$ (138) = 2.35, $p = 0.02$]). In addition, the number of EEG recordings contributed per participant did not appreciably impact the EEG power estimates derived from the longitudinal data ($p > 0.05$ in 35 of 36 Pearson's correlations between number of EEG recordings per participant and participants' EEG power parameters in primary analyses over frontal cortex; number of recordings was weakly associated with theta band 12-month power in the 12–24 month frontal analysis only [Pearson's correlation $r = 0.183$, $p = 0.045$]).

| Table 5 Sample demographics | | | |
|---|---|---|---|
| | **Low-risk control** $n = 69$ | **High-risk No ASD** $n = 71$ | **High-risk ASD** $n = 31$ |
| Sex | 37 M, 32 F | 34 M, 37 F | 22 M, 9 F |
| *Child ethnicity (%)* | | | |
| Caucasian | 85.5 | 93.0 | 80.6 |
| Hispanic/Latinx | 1.4 | 4.2 | 16.1 |
| Asian American | 2.9 | 2.8 | 6.5 |
| African American | 1.4 | 1.4 | 0 |
| Multirace | 8.7 | 2.8 | 12.9 |
| Mean household income ($1000 s) | 65–75 | 65–75 | 65–75 |
| *Mean parental education (%)* | | | |
| <4 year college | 4.3 | 16.9 | 19.4 |
| =4 year college | 16.0 | 19.7 | 29.0 |
| >4 year college | 70.0 | 54.9 | 32.3 |
| Included EEG data | | | |
| *By visit age (n)* | | | |
| 3 months | 10 | 15 | 9 |
| 6 months | 51 | 39 | 15 |
| 9 months | 55 | 49 | 21 |
| 12 months | 61 | 45 | 26 |
| 18 months | 45 | 48 | 22 |
| 24 months | 46 | 48 | 20 |
| 36 months | 49 | 45 | 16 |
| *By analysis (n)* | | | |
| 3–12 months | 63 | 51 | 22 |
| 12–24 months | 54 | 48 | 25 |
| 3–36 months | 69 | 71 | 31 |
| In all analyses | 52 | 38 | 20 |
| *By participant (mean number of EEG recordings [SD])* | | | |
| 3–12 months analysis | 2.7 (0.64) | 2.6 (0.64) | 2.9 (0.77) |
| 12–24 months analysis | 2.5 (0.50) | 2.5 (0.51) | 2.4 (0.49) |
| 3–36 months analysis | 4.6 (1.22) | 4.1 (1.39) | 4.2 (1.55) |
| *Included EEG HAPPE metrics (mean [SD] range)* | | | |
| Length of raw EEG (seconds) | 169.8 (94.3) 44–784 | 182.9 (128) 44–1067 | 191.1 (105) 64–594 |
| Good channels (%) | 91.9 (4.5) 82.1–100 | 92.9 (4.5) 82.1–100 | 91.9 (4.3) 82.1–100 |
| Rejected components (%) | 41.3 (11.8) 0–68.8 | 42.0 (12.7) 5.1–74.3 | 38.9 (15.0) 0–72.2 |
| EEG variance retained (%) | 63.6 (14.9) 32.1–100 | 63.7 (15.3) 33.6–98.1 | 68.3 (14.5) 39–100 |
| Mean retained artifact probability | 0.16 (0.05) 0.03–0.30 | 0.17 (0.04) 0.047–0.28 | 0.16 (0.05) 0.047–0.25 |
| Median retained artifact probability | 0.13 (0.08) 0.01–0.34 | 0.13 (0.06) 0.01–0.31 | 0.12 (0.06) 0.01–0.25 |
| EEG segments retained (n) | 74.6 (40.7) 21–334 | 81.0 (54.7) 21–474 | 84.3 (46.5) 31–260 |
| *SD* standard deviation | | | |

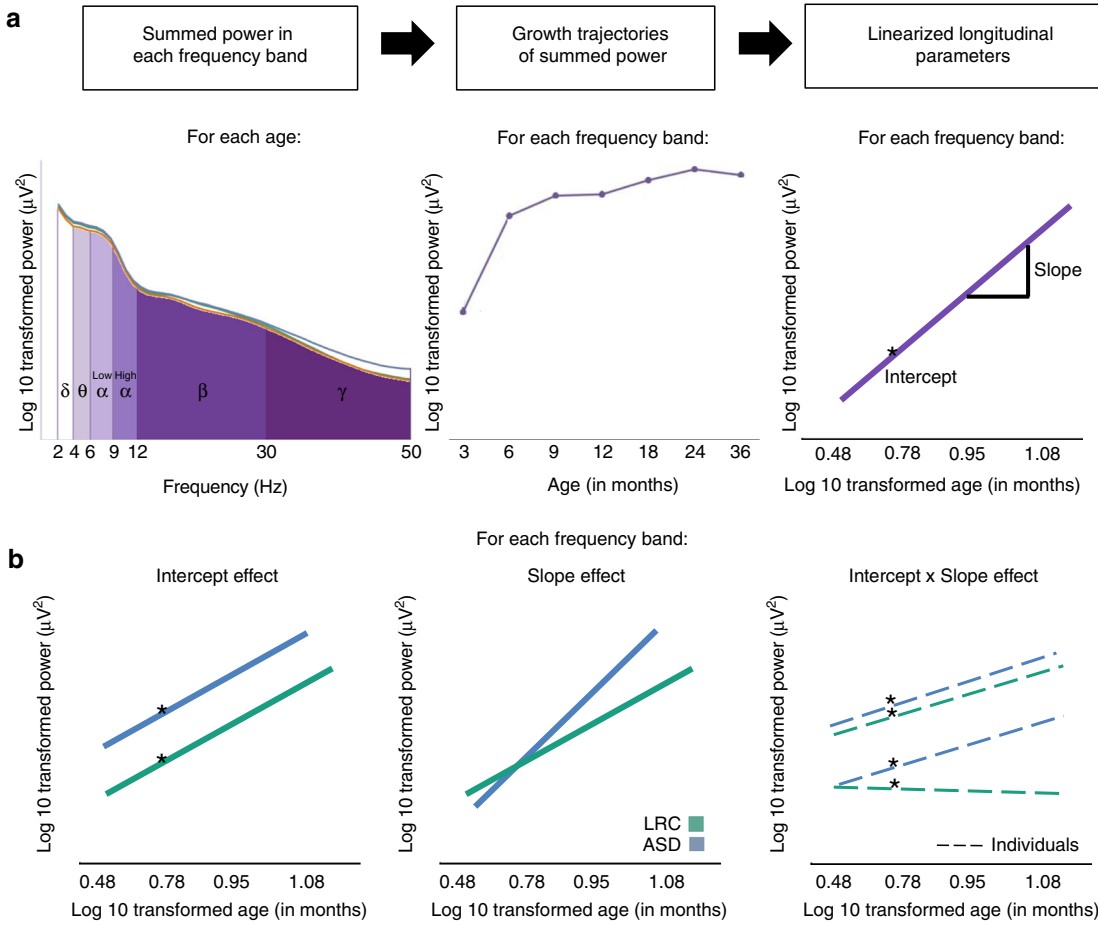

**Fig. 2** Analysis schematic. Conceptual diagram illustrating how longitudinal EEG parameters were generated and analyzed. **a** (For each participant): at every age, the total, summed power in each canonical frequency band (delta, theta, low alpha, high alpha, beta, gamma labeled with their Greek symbol equivalents) was calculated as the area under the curve of the EEG power distribution (left panel). Growth trajectories of summed power in each frequency band were generated across (1) 3–12, (2) 12–24, and (3) 3–36 postnatal months of age (beta frequency band 3–36 month trajectories plotted here, middle panel). Growth trajectories of summed power from each frequency band were linearized by modeling log (summed power) as a function of log (age) for each frequency band; this allowed for the calculation of an estimated intercept (here, at 6 months) and a linear developmental slope for each individual with at least two EEG recordings to be submitted to group-level analysis (right panel, here for beta frequency band). **b** (Group-level effects): three types of effects were tested for in the data-driven model construction to differentiate all pairs of groups (here, the low-risk control (LRC, in green color) vs. Autism (ASD, in blue color) group comparison is shown): main effects of differences in intercepts (left panel), and developmental slopes (middle panel), and interaction effects between intercept and slope (right panel). The interaction effect tested whether the relation between intercept and slope varied between groups (e.g., here, individuals in the ASD group have the same slope regardless of having low or high intercept values, but individuals in the LRC group have steeper slope values with higher intercept values)

**EEG data acquisition.** Baseline, resting EEG data were acquired while the infants were seated on their caregivers' laps in a dimly lit, sound-attenuated, electrically shielded room. Continuous EEG was recorded for between approximately 2 and 5 min. A research assistant sat to the side of the caregiver and infant. They were in the room to assist in keeping the infant calm and still by blowing bubbles across the room or presenting a quiet toy to the infant if they became fussy (e.g., a ball). Research assistants did not engage the infants in social interaction. This EEG collection environment follows both standard empirical study conditions for baseline EEG collection (e.g., [48] and serves as a scalable, naturalistic EEG paradigm that can translate to clinical settings. EEG data were collected with Geodesic Sensor Nets, using a 0.1 Hz high-pass analog (i.e., hardware) filter and online rereferencing to the vertex (channel Cz) through NetStation software (Electrical Geodesics, Inc (EGI), Eugene, OR, USA). Impedances were kept below 100KΩ in accordance with the impedance capabilities of the high-impedance amplifiers inside the electrically shielded room[49].

**EEG processing.** The continuous, baseline EEG data were first exported from NetStation native format to MATLAB format (R2017a). Data preprocessing, artifact removal, and data quality assessment was carried out via the Harvard Automated Processing Pipeline for EEG (HAPPE), a preprocessing pipeline optimized for developmental EEG data[50]. HAPPE has been shown to reject more artifacts and simultaneously preserve more EEG signal during processing than other

contemporary EEG processing approaches, including manual editing[50]. All files were batch processed using the batch electroencephalography automated processing platform (BEAPP) software, allowing for the same empirical criteria for artifact removal to be applied to each file in the same way[51]. To optimize artifact rejection performance given the lengths and sampling rates in the EEG data, a spatially distributed subset of channels providing whole-head coverage was processed through HAPPE (64-channel net—2, 3, 6, 8, 9, 11, 12, 13, 15, 16, 17, 21, 24, 25, 27, 28, 34, 37, 40, 46, 49, 50, 52, 53, 54, 57, 58, 61, 62; 128-channel net—3, 4, 9, 11, 13, 19, 20, 22, 23, 24, 27, 28, 33, 36, 40, 41, 45, 46, 47, 52, 58, 62, 70, 75, 83, 92, 96, 98, 102, 103, 104, 108, 109, 112, 117, 118, 122, 123, 124; Supplementary Fig. 1). For each EEG, a 1 Hz digital high-pass filter and a 100 Hz low-pass filter was applied in preparation for independent component analysis[52]. Data sampled at 500 Hz were then resampled with interpolation to 250 Hz as recommended for HAPPE processing (resampling was performed after filtering to avoid aliasing higher frequencies when resampling). HAPPE's artifact removal steps included removal of 60 Hz electrical noise through CleanLine's multi-taper approach[53], bad channel rejection, and participant artifact rejection (e.g., eye blinks, movement, and muscle activity) through wavelet-enhanced ICA with automated component rejection via EEGLAB[54,55] and the Multiple Artifact Rejection Algorithm[56]. Post-artifact rejection, any channels removed during the bad channel rejection were repopulated through spherical interpolation to reduce spatial bias in rereferencing. The EEG data were then rereferenced to the average reference and mean signal detrended. The EEG was segmented into contiguous 2-s windows and any segments with

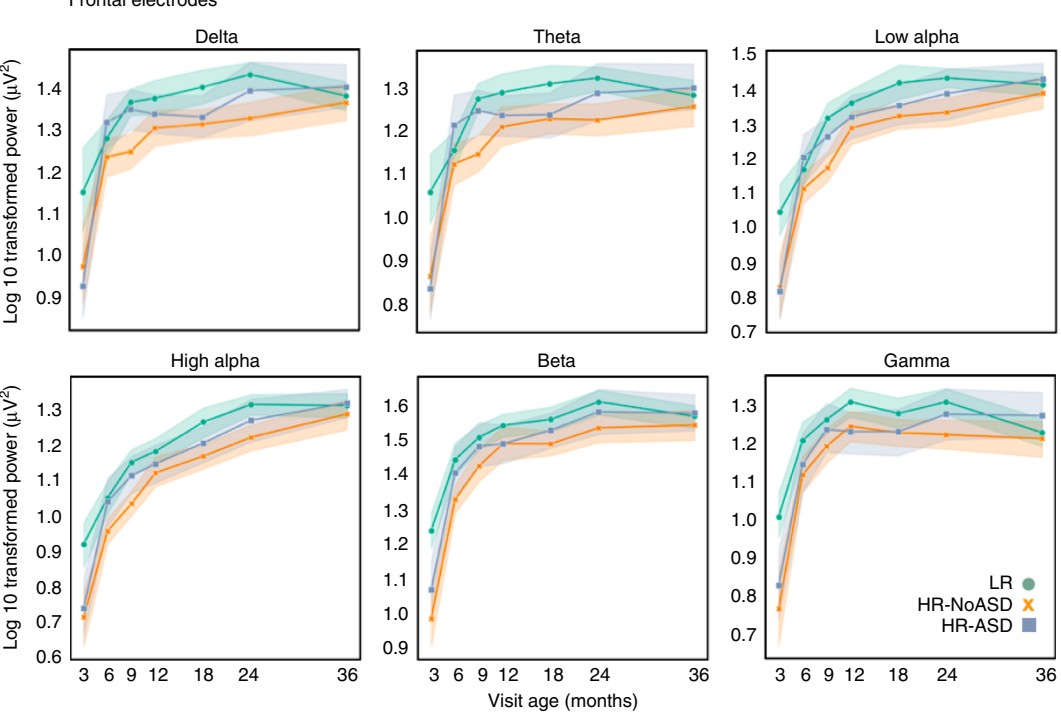

**Fig. 3** EEG power from 3 to 36 months of age. Mean log of EEG power, calculated as the sum of the power across each frequency band, is shown for each frequency band over all visit ages for each outcome group, low-risk control (LRC, represented by green circles), high-risk without Autism (HRA−, represented by orange letter x), and high-risk with Autism (ASD, represented by blue squares) for the frontal region of interest. Lines connecting power values across visit ages are to aid visualization. Error bars are 95% confidence intervals around the mean

retained artifact were rejected using HAPPE's amplitude and joint probability criteria, as in prior research with developmental EEG[57]. Noncontiguous data were not concatenated during this process. Importantly, outcome groups did not significantly differ in the mean lengths of their processed EEG or in any of the HAPE data quality measures (all Student's $t$ test $p > 0.05$, Table 5).

**EEG rejection criteria.** Standard HAPPE processing includes the application of recommended thresholds to HAPPE data quality metrics to identify and reject problematic EEG files postautomated processing. These quantitative file quality thresholds together with the automated data rejection procedures within HAPPE facilitate reproducible analyses. EEGs were rejected if they had fewer than 20 postprocessed good segments (40 total seconds of EEG) in keeping with recommendations for achieving highly reliable EEG power estimates[39,40], or were more than 3 standard deviations (SD) from the mean on the following HAPPE metrics: percent good channels (3SD: <82%), mean retained artifact probability (3SD: >0.3; the estimated probability that retained independent components contain any artifact, calculated as an intermediate processing metric before further segment rejection based on remaining artifact), median retained artifact probability (3 SD: >0.35; the median value for the same artifact probability metric as the mean values), percent of independent components rejected as artifact (3SD: >84%), and percent of EEG signal variance retained after artifact removal (3SD: <32%). EEGs with a mean power greater or less than two SD from their outcome group for each frequency band were visually reviewed blind to outcome group status. This led to the rejection of 25 additional EEG recordings (9 LRC EEGs, 11 HRA- EEGs, 5 ASD EEGs) out of 760 recordings (3% of recordings) and only 1 participant to be dropped from analysis. HAPPE quality metrics and visual inspection rejection rates did not significantly differ between any pair of outcome groups (all Student's $t$ test $p > 0.1$, Table 5). Supplemental Pearson correlation analyses assessed whether the lengths of either the raw or post-HAPPE processed EEG files were associated with the amount of artifact in the data (measured directly by the percent of independent components rejected and the percent of variance retained after artifact removal). There were no significant associations between the average lengths of the raw or processed EEG per participant and the amount of artifact detected in the EEG data across the 3–12, 12–24, or 3–36 month analyses (all Pearson's correlation $p > 0.10$).

**EEG power decomposition.** A Fast Fourier Transform with multitaper windowing (3 tapers) was used to decompose the EEG signal into power for each 2-s segment for each of the channels of interest. Two nets (64-channel and 128-channel) were used across the course of the longitudinal study (the switch from 64- to 128-

channel nets occurred because the company ceased supporting the 64-channel net equipment), so channels were selected for analysis from spatial locations that corresponded across nets (EGI, Eugene, OR). Channels clustered over the frontal cortex were analyzed in primary analyses (64-channel net—2, 3, 8, 9, 12, 13, 58, 62; 128-channel net—3, 4, 11, 19, 20, 23, 24, 27, 118, 123, 124; Supplementary Fig. 1). Supplemental analyses were performed to compare the results from the a priori selected frontal cortex region of interest with both a distributed, standard layout and an additional clustered region of interest. These supplemental layouts consisted of HAPPE's whole-head 10–20 channel equivalents (64-channel net—3 (Fz), 13 (F3), 62 (F4), 15 (F7), 61 (F8), 17 (C3), 54 (C4), 24 (T3), 52 (T4), 27 (T5), 49 (T6), 28 (P3), 34 (Pz), 46 (P4), 37 (O1), 40 (O2); 128-channel net—11 (Fz), 24 (F3), 124 (F4), 33 (F7), 122 (F8), 36 (C3), 104 (C4), 45 (T3), 108 (T4), 58 (T5), 96 (T6), 52 (P3), 62 (Pz), 92 (P4), 70 (O1), 83 (O2)), and channels clustered over bilateral temporal-parietal cortex (64-channel net—21, 24, 25, 28, 46, 50, 52, 53; 128-channel net—40, 41, 45, 46, 47, 52, 92, 98, 102, 103, 108, 109).

For each of the six frequency bands typically used in infant EEG studies, the summed power across all frequencies within the band was calculated as the measure of total power in that frequency band (i.e., power density, or power/Hz was not used in these analyses). Here, summed power in a frequency band is equivalent to the area under the power density curve for that frequency band; Fig. 2). Frequency bands were defined as follows: delta, 2–3.99 Hz; theta, 4–5.99 Hz; low alpha, 6–8.99 Hz; high alpha, 9–12.99 Hz; beta, 13–29.99 Hz; gamma, 30–50 Hz. For each EEG recording and each channel, the average summed power in each frequency band across all 2-second segments was calculated and normalized by a log base 10 transform. For each EEG recording, the summed power in each frequency band was then averaged across all channels of interest. (See Fig. 3 for the mean EEG summed power in each frequency band at each timepoint for the frontal configuration used in the primary analyses).

**Longitudinal EEG power parameters (within-participant).** Within-participant analyses were carried out in SAS software. To reduce the number of EEG parameters tested in the primary statistical analyses differentiating outcome and risk groups, the trajectories of EEG summed power within each frequency band across development were summarized by two parameters per frequency band: EEG summed power intercepts and the slope of EEG summed power across age (developmental slope; Fig. 2). To generate these parameters, we ran ordinary least squares (OLS) regressions for each infant, modeling that infant's age-related change in EEG summed power within each frequency band. The OLS regressions modeled log-transformed EEG summed power as a function of log visit age, given the nonlinear, logarithmic age-related changes in log-transformed EEG summed

power observed across outcome groups and frequency bands. Modeling log-EEG summed power as a function of log-age linearized the relation between EEG and age (i.e., a linear regression line could be fit between these log-transformed variables). Infants therefore needed EEG recordings from only two visits to calculate this linear regression line, and contribute data to between-participant analyses.

Individual OLS regressions were run using all available data for each infant across three different age ranges: 3–12, 12–24, and 3–36 months of age. The 12-month time point was included in both the 3–12 and 12–24 month windows to maximize the sample size of infants contributing to both sets of analyses, and to facilitate more complete coverage of the first and second postnatal years in each respective analysis.

For each developmental window, participants' OLS regression equations were used to generate their individual summed power intercept and linear developmental slope values in each frequency band for between-participant analyses. For the summed power intercept values in the 3–12 and 3–36 month analyses, the estimated summed power at 6-months of age was calculated from the fitted regression equation for each infant (including infants who had 3-month data). We chose the estimated summed power at 6 months of age rather than 3 months of age for the initial value so we could more easily compare trends at 6-months of age to prior studies whose earliest timepoint was 6 months of age, and because more infants across all risk and outcome groups contributed EEG data at the 6-month timepoint than at the 3-month timepoint (which was added later in the study). In the 12–24 month EEG analyses, 12 months was used for the EEG summed power intercepts for all infants. EEG summed power developmental slopes were calculated as the linear slope parameter estimates from the OLS regression model over each developmental window.

**Statistical software**. All analyses were carried out in R software[58]. All reported statistical tests in the present study are two-sided tests wherever applicable.

**Data-driven model construction**. Potential predictors of interest included the EEG summed power intercepts, the EEG summed power developmental slopes, and parameters modeling the interaction terms between the intercept and slope within each frequency band for the entire set of frequency bands (18 total parameters; Fig. 2). Given that child sex and parental education differed between groups, these two variables were also entered as potential predictors to ensure that the EEG parameters were differentiating outcome groups over and above the group differences in sex and parental education.

Binary logistic regression models were constructed to understand the predictive capabilities of EEG parameters adjusting for sex and parental education. Model selection was carried out with a hybrid stepwise selection approach (i.e., data-driven) by minimizing the Akaike Information Criteria (AIC). AIC provides a relative estimate of a model's out-of-sample predictive accuracy, where lower AIC values indicate more accurate models (that is, AIC penalizes over-fitting to the current sample)[59–62]. However, for sparse data, like the ASD group in the present study, AIC is a pragmatic method to avoid parsing small samples[60,63]. Here, AIC was used to identify the most accurate, generalizable model generated by the data that differentiated outcome groups without overfitting to the present sample[59]. Primary analyses modeled three group comparisons separately as these are not collectively exhaustive categories: the ASD vs. HRA− groups, the ASD vs. LRC groups, and the HRA− vs. LRC groups. Binary logistic regression models were each constructed using (a) the EEG data measured between 3 and 12 months, (b) EEG data measured between 12 and 24 months, and (c) EEG data from 3 to 36 months. Once the data-driven model selection process was completed, post hoc hypothesis testing (Student's t-tests on parameter estimates within the multiple logistic regression) was performed to identify the relatively robust parameters that differentiated outcomes in each model.

**Model comparison criteria**. To compare the AIC-minimized logistic regression models' performances across developmental windows and between group comparisons, in-sample receiver-operating characteristic (ROC) curve criteria were used[64]. First, to compare how accurately the logistic models differentiated between outcome groups, the area under the ROC curve (AUC) was calculated for each model with bootstrapped 95% confidence intervals. The AUC metric is robust under conditions of skewed sample sizes between groups, as in these data. The sensitivity and specificity of the models were then evaluated using the threshold that maximized Youden's Index (i.e., Youden's J statistic) to optimize the model's combined sensitivity and specificity rates[65]. To assess the true positive and true negative differentiation rates across models, we calculated the PPV and NPV, respectively at the threshold corresponding to Youden's Index. These predictive values account for the skewed disorder prevalence in the sample[64].

**Reporting summary**. Further information on research design is available in the Nature Research Reporting Summary linked to this article.

## Data availability

The datasets analyzed during the current study are available from the corresponding author on reasonable request.

## Code availability

The statistical analysis code used during the current study is available from the corresponding author on reasonable request. The code to process the EEG data is freely available under the HAPPE and BEAPP software licenses (HAPPE: https://github.com/lcnhappe/happe; BEAPP: https://github.com/lcnbeapp/beapp).

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

## Acknowledgements

The present research was supported by the Autism Science Foundation (to L.G.-D., C.L.W. and A.L.), the Rett Syndrome Research Foundation (to L.G.-D.), the University of Tokyo International Research Center for Neurointelligence (to L.G.-D), the American Brain Foundation (to A.L.), the Nancy Lurie Marks Family Foundation (to A.L.), the Brain and Behavior Research Foundation (to A.L.), the FRAXA Research Foundation (to C.L.W.), and the National Institutes of Health (1T32MH112510 to C.W. and R01-DC010290 to H.T.F. and C.A.N.).

## Author contributions

C.A.N. and H.T.F. designed the longitudinal study. L.G.-D., C.W., and A.L. processed the electrophysiological data. L.G.-D., C.W., K.K., and A.L. conceived and carried out the analyses. L.G.-D. drafted the paper, and all authors provided critical revisions. All authors approved the final version of the paper for submission.

## Additional information

**Competing interests:** The authors declare no competing interests.

