## [Transparent Peer Review File · Nature Communications]

Reviewers' comments:

Reviewer #1 (Remarks to the Author):

This interesting paper reports EEG power analyses from a longitudinal sample of infants and children to test for differences in EEG as a function of ASD risk. The findings were novel in that developmental trajectories of EEG power were shown to be related to risk status—as opposed to measures taken at a single age, as in much extant research using similar methods. I thought the paper was very compelling but only somewhat convincing at present; the measures were not described in sufficient detail, as elaborated subsequently. Specific comments and questions:

Title, and elsewhere in the ms: “after birth” is preferable to “of life,” as the latter implies that development begins at birth.

lines 35-36: EEG can be measured in fetuses (e.g., 10.1016/j.brainres.2017.10.010)

lines 73-75: What kinds of developmental change are found in frontal EEG?

lines 96-98, etc.: I never was quite clear why frontal EEG was analyzed for all three time windows, but whole-head EEG was analyzed only for the youngest window.

lines 150-153: It isn't clear from this section or from Table 1 how many infants contributed EEG data at any particular time point. The ms notes that EEG was recorded at 3, 6, 9, 12, 18, 24, and 36 months, but not how many EEG recordings were actually included in the analyses. (Table 1 notes how many children were included, but not how many recordings.) Nor is it clear whether the 12-month recordings were included in both the 3-12 month analyses and the 12-24 month analyses (cf. lines 252-256).

lines 178-181: How does BEAPP software “ensure uniform analysis on the EEG data occurred regardless of when the infant was tested and what study group the infant was assigned?”

line 191: “was” —> “were”

lines 255-264: I found this section confusing because (as noted before) it isn't clear how much data were contributed by each infant; moreover I couldn't follow the discussion of the 6-month intercept (or put another way, how 3-month-olds' data were handled).

lines 320-361: Do these results concern total EEG power, i.e. collapsed across all frequency bands?

lines 369-370: What about parents' education? Is that important?

Table 1: Earlier (line 229), it is noted that EEG samples must be 40 s to be included in the data set. The means for these values in Table 1 for each group are considerably longer than this, with large SDs. I wondered if the ranges should be included here also, and I was concerned about differences in data quality between short and long samples. Also, what is “Mean retained artifact probability?” (and median)

Figure 1: I struggled with Figure 1. It wasn't clear to me how to relate what is plotted here with the rest of the results. The most salient feature of the three plots (I thought) is the low power for all frequency bands in the HRA- and ASD groups at the 3-month time point,. However, it is mentioned (lines 257-258) that not all infants contributed 3-month data, and the number of infants is not provided—perhaps this jump is an artifact? Does it mean something? The other slopes are hard to interpret and there's not a clear pattern other than obvious differences in power between different bands in all three groups. Nevertheless, the 3-12 month slope in delta and theta

bands (Table 3) assume importance in the overall interpretation of the study. Thus it may be important to describe what data are included in the analyses (echoing previous comments).

Scott P. Johnson, UCLA (I sign all reviews)

Reviewer #2 (Remarks to the Author):

The manuscript reports results of a study aiming to identify infant resting EEG power markers predictive of ASD-related outcomes. Given the benefits of early interventions, there is a high need for an objective indicator that could be used to reliably diagnose or identify risk for ASD prior to a behavioral diagnosis. This study represents the next step in the ongoing work by this group (e.g., Tierney et al., 2012; Levin et al., 2017) and now includes the 36-month data and diagnostic evaluation results that allowed to stratify the original infant sample into the ASD, high-risk, and low-risk subgroups.

The novel aspects of this longitudinal study conducted in a large sample of infant siblings include a systematic evaluation of the resting EEG diagnostic predictive power at various developmental points between 3-36 months, as well as the quantification of the developmental trajectories for the EEG power metrics and an assessment of their sensitivity to clinical (ASD vs. others) and subclinical (high-risk vs. low risk) features. The authors report that the frontal EEG power during 3-12 month was the best at discriminating the ASD group from the others, while the combined data across 3-36 months discriminated the high-risk from low-risk groups. Analyses of the individual frequency bands identified multiple low- and high-frequency power metrics at various developmental stages that could differentiate the three subgroups.

The reported findings provide new support for the suggestion made by this and other groups (e.g., Wang et al., 2015 J of Neurodevelopmental Disorders) that differences in resting EEG characteristics may reflect ASD pathophysiology, and highlight the importance of considering a developmental course (trajectory over time). The paper will be of interest to the field of autism in general and to research teams working on neurophysiological markers of ASD for diagnostic or sample stratification and treatment evaluation purposes.

My main concern is with the use of the term "biomarker." Increasingly, studies label any physiological measures sensitive to group differences as biomarkers without explicitly demonstrating that they indeed meet the criteria. The retrospective design is helpful as the first step for identifying candidate biomarkers for further testing. However, and the authors acknowledge this in the limitations, they need to be validated in new samples and ideally, by other, independent teams. In the current study, 18 different frontal EEG power measures (6 frequency bands x 3 metric types) were sensitive to some diagnostic group differences in at least one of the reported analyses. Which of these measures (or a combination of them) would the authors recommend for further, more detailed validation?

To be clinically meaningful, a biomarker needs to be specific to the diagnostic condition. In the current paper, the multiple bands that differentiated ASD and/or high-risk from low-risk infants would reflect a variety of neural processes (the authors noted inhibition and experience-dependent plasticity as some of the possibilities). Could the authors comment on how one would determine whether the observed resting EEG power differences are specific to ASD or reflect more general language difficulties, executive function deficits, etc.?

On a related note, the manuscript suggests that frontal EEG power is the most informative marker, but the rationale for initially selecting this location was not clearly established. Furthermore, the empirical evaluation of spatial specificity might have been confounded by differences in signal quality. The frontal EEG power metrics came from a large cluster of spatially contiguous electrodes with highly correlated data, while the whole-head dataset was an average of spatially distributed

single channels. Thus, the frontal EEG likely had better signal-to-noise ratio, which could explain greater sensitivity to the diagnostic group differences compared to the whole-head average. The goal of evaluating spatial specificity of the frontal EEG biomarkers could have been tested more directly if the frontal cluster data were compared to similar-sized clusters in temporal, parietal, or occipital regions.

A biomarker, especially a diagnostic one, also needs to be reproducible: the current study did not include a test of reliability of the EEG power measures. Different EEG frequency bands and power metrics (intercept or slope) at different ages were predictive of the later diagnostic outcomes, with some losing their predictive value over time. The authors attributed this potential instability to extensive neural development in the first three years of life, but it is unclear whether the underlying pathophysiology of ASD reflected by the individual frequency bands would have changed significantly from 6 to 12 to 24 months, etc.

Other comments:

The frontal electrodes used in the analysis appear to represent two lateralized clusters with a narrow midline "connector" (also, Supplementary Figure 1 may have an incorrect color code for channel 3 in the 64-channel net). Previously, this group (Gabard-Durnam et al., 2015) suggested that hemisphere asymmetry in frontal EEG may reflect an autism endophenotype in infants. Thus, it would be interesting to know why hemisphere differences were not considered in the present study.

Previously, Tierney et al. (2012) suggested that age-dependent group differences in EEG power may be less informative than developmental trajectories of change. Using the current large data set, could the authors comment on whether the slope of EEG power is indeed a more robust marker than EEG power at a single age?

The biggest challenge to using EEG as a diagnostic biomarker to date has been the inability to interpret the data at the individual level. This is a major difficulty for the field of human neurophysiology because the observed EEG signal may vary based on equipment, acquisition and signal processing routines that tend to be different across labs. Given the authors' expertise and interest in biomarker development, could they comment on the next steps toward moving from group-level to individual-level results?

Reviewer #3 (Remarks to the Author):

This study represents a large longitudinal examination of resting EEG developmental trajectories as potential early discriminative biomarkers for ASD and risk for ASD. While the study has the potential to be a foundational piece in the literature there are some methodological concerns that reduce the generalizability and potentially also the reliability of the results.

Introduction is well written and straightforward.

Method

Please list maximum number of channels rejected with BEAPP (this is not automatically regulated by beapp as far as I know so can be a concern). Was HAPPE done on all 64/128 channels or just the ones used in the analyses? If just on the reduced number of electrodes, then rejecting/interpolating more than a single channel is problematic (usually 5% or fewer interpolated channels is a general guideline), especially given the referencing to average reference.

Resampling to 250 Hz and low passing at 100 Hz violates the Nyquist rule (as it is generally applied to EEG) where the anti-aliasing low pass filter should be less than 1/3 of sampling frequency (so low pass should be somewhere around 80 Hz maximum for a 250 Hz sampling rate).

This could impact gamma and beta findings if high gamma gets aliased to beta, which could be happening given the large amount of beta that is apparently occurring in this data.

Impedances at 100 kohm is pretty high. The conservative standard for EGI is 50 kohm with 75 being a more liberal threshold. 50 kohm should be easily attainable with infant scalp. Is there a reason why impedances were allowed to be so high?

Results

Statistical methods appear to be appropriate and thorough.

The results section states the frequency bands with significant differences between groups, but perhaps a bit more description is warranted here on how they differ (e.g. gamma decreases in one group over time but not another), rather than waiting until the discussion section for this.

Discussion

Discussion says that frontal electrodes results are better than whole head but the results say findings were largely the same. Please reconcile.

Figures

I have some concerns about the power results in figure 1. Power usually follows a $1/f$ distribution, with low frequencies having highest power yet the log power in figure 1 shows relatively low power in low frequencies and very high power in beta band, which is not consistent with any other developmental studies that I am aware of. Residual motion artifact might explain the increase in beta band power, or aliasing from gamma. Alternatively several infant EEG papers recently published by some of these coauthors and presumably collected in the same lab set-up show this increase in beta relative to the other frequencies, which may point to something that may not generalize to other research groups. Although the groups did not differ on artifact rejection criteria via HAPPE, it is important to identify the source (theoretical neural or artifactual) for this unusual finding in case different groups have different contributing factors to this beta increase (i.e. one is more prone to certain types of artifact than the other at different developmental windows, contributing to group trajectory differences in specific frequency bands). Do the authors have a mechanistic theory for why the power spectra are shaped this way? Tierney 2012, reporting over similar age range and electrodes, shows a standard $1/f$ distribution in power, as well as a decrease in beta and gamma power with age that is dissimilar to what is reported here. Given the limited number of studies available for comparison, how might we reconcile these findings?

Separating the power trajectory plots by group in Figure 1 and presenting all the frequency bands together on one plot lessens the ability to distinguish between group differences by frequency band. As presented, the small differences in trajectory are difficult to decipher beyond the general impression that they all look quite similar across groups. It might be more beneficial to present a plot for each frequency band with different lines for each group in order to make more direct comparisons for intercept and trajectory (similar to the Tierney paper referenced above).

We thank each of the reviewers for their very thoughtful, constructive comments on the original manuscript. Below we note the edits and changes we have made in response to each reviewer's concerns. Please note that we address the concern about EEG power distributions raised by Reviewer #3 and the editor in response to comment # 27 within this document.

Reviewer #1 (Remarks to the Author):

1. Title, and elsewhere in the ms: "after birth" is preferable to "of life," as the latter implies that development begins at birth.

We thank the reviewer for catching this point, and we agree. We have changed each instance referencing "years of life" to "after birth" or "postnatally" in the manuscript body and title.

2. lines 35-36: EEG can be measured in fetuses (e.g., 10.1016/j.brainres.2017.10.010)

We thank the reviewer for this correction. We have edited those lines in the abstract to read:

Lines 34-35:

"Brain oscillations captured in electroencephalography (EEG) are measurable from prenatal ages onwards and are..."

3. lines 73-75: What kinds of developmental change are found in frontal EEG?

We have added more text to the introduction to describe the prior findings with frontal EEG in development (see excerpt below).

Lines 71-84:

"In particular, differences between ASD risk groups in EEG spectral power across a range of frequency bands over the frontal cortex have been shown to emerge early in infancy and relate to later symptomatology (9–16). Infants with elevated risk of ASD have been shown to have lower frontal alpha and beta band power at 3 months of age, lower frontal power from delta through gamma frequency bands by 6 months of age, and different age-related changes in delta, alpha, beta, and gamma band power through two years of age, showing largely steeper age-related increases in power (10, 11, 17). Compared to other cortical regions, frontal EEG power consistently demonstrates differences in theta, alpha, and gamma bands during the second year postnatally that have been associated with a range of ASD symptom domains in this elevated-risk population, including sensory hyporesponsiveness, cognitive deficits, language, and the degree of restricted and repetitive behaviors (10, 12–17). However, it remains to be determined how well early EEG differences can distinguish subsequent ASD outcomes."

Lines 99 – 103:

“Here, we leveraged longitudinal baseline EEG in a cohort of these high familial risk infant siblings and infants at low familial risk for ASD to derive data-driven profiles of EEG power across multiple timescales. Given the robust early differences observed over frontal cortex, the present study focused a priori on frontal EEG power to address this question.”

4. lines 96-98, etc.: I never was quite clear why frontal EEG was analyzed for all three time windows, but whole-head EEG was analyzed only for the youngest window.

We apologize for the lack of clarity in our analysis plan. Our primary goal was to identify the developmental window when frontal EEG power best discriminated outcome groups, as the majority of prior work on early EEG differences has found or focused frontal power differences. Once we had identified this window with frontal electrodes, we then planned to examine the 10-20 configuration of electrodes to see whether the frontal findings were specific to frontal electrodes or generalized across the scalp. Given the high performance of the frontal electrode 3-12 month model, we performed this second analysis with the 10-20 electrodes in the same 3-12 month window.

However, we agree there is utility in presenting the complete set of analyses for the 10-20 electrodes to allow for comparisons between spatial layouts across the full three year period of interest. Therefore, we have performed the 12-24 months and 3 – 36 months analyses for the 10-20 electrode layout. We have edited Table 2 to include the model performance measures (e.g. accuracy, sensitivity, etc) for all 10-20 models. To focus the main text on the parameters from the a priori frontal region of interest, we have included the parameters for the 10-20 electrode models in the supplemental materials for the revised manuscript.

Please note that Reviewer 2 inquired about comparisons between frontal electrodes and other spatially-localized regions of interest (please see comment and response #17 below), so we have now also examined model performance for a bilateral temporal-parietal region of interest over these same developmental windows. (Notably, frontal EEG power over the first postnatal year still provides the most accurate differentiation of the infants who later receive an ASD diagnosis.)

5. lines 150-153: It isn't clear from this section or from Table 1 how many infants contributed EEG data at any particular time point. The ms notes that EEG was recorded at 3, 6, 9, 12, 18, 24, and 36 months, but not how many EEG recordings were actually included in the analyses. (Table 1 notes how many children were included, but not how many recordings.)

We agree with the reviewer that we did not provide enough information about the longitudinal EEG recordings in the manuscript, and we have significantly edited Table 1 and the methods section in the revised manuscript to address these points. We now include in Table 1 both the sample size for each outcome group at each age, and the mean number of EEG recordings per participant in each outcome

group for each analysis (These Table 1 additions are replicated below for convenience). We have also performed several additional analyses to check that the number of contributed EEG recordings per participant did not influence our analyses. First, the number of EEG recordings contributed per participant did not differ between outcome groups for all but one outcome group pair in one analysis. Moreover, the number of EEG recordings contributed did not correlate with the EEG power parameter estimates for the overwhelming majority of parameters across layouts and developmental windows (71 of 72 correlations for frontal and 10-20 parameters returned $p > 0.05$, except theta 12-month intercept parameter in the 12-24 month frontal analysis [Pearson's $r = 0.183$, $p = 0.045$]). Therefore, variance in the number of EEG recordings that participants contributed does not appear to unduly affect our analyses.

Revised Table 1 section on EEGs included:

Included EEG data:	LRC	HRA-	ASD
By visit age (n)			
3 months	10	15	9
6 months	51	39	15
9 months	55	49	21
12 months	61	45	26
18 months	45	48	22
24 months	46	48	20
36 months	49	45	16
By analysis (n)			
3 - 12 months	63	51	22
12 - 24 months	54	48	25
3 - 36 months	69	71	31
In all analyses	52	38	20
By participant (mean number of EEG Recordings [SD])			
3 - 12 months analysis	2.7 (0.64)	2.6 (0.64)	2.9 (0.77)
12 - 24 months analysis	2.5 (0.50)	2.5 (0.51)	2.4 (0.49)
3 - 36 months analysis	4.6 (1.22)	4.1 (1.39)	4.2 (1.55)

Revised methods text lines 163-180:

“EEG data were collected when infants were 3, 6, 9, 12, 18, 24, and 36 months of age. Not all infants contributed adequate EEG data at all timepoints. The sample size of usable EEG for each outcome group at each timepoint, sample sizes for each of the primary analyses run across timepoints, and the mean number of EEG recordings that each participant within each outcome group contributed to primary analyses are provided in Table 1. Across outcome groups, comparable numbers of EEG recordings were contributed per participant. For primary analyses (3-12 months, 12-24 months,

and 3-36 months) the mean number of EEG recordings contributed per participant did not significantly differ between pairs of outcome groups in 8 of the 9 comparisons (only LRC and HRA- groups in the 3-36 month analysis significantly differed, with LRC infants contributing more EEGs [$t(138) = 2.35, p = 0.02$]). Additionally, the number of EEG recordings contributed per participant did not appreciably impact the EEG power estimates derived from the longitudinal data ($p > 0.05$ in 35 of 36 Pearson's correlations between number of EEG recordings per participant and participants' EEG power parameters in primary analyses over frontal cortex; number of recordings was weakly associated with theta band 12-month power in the 12-24 month frontal analysis only [Pearson's $r = 0.183, p = 0.045$])."

6. Nor is it clear whether the 12-month recordings were included in both the 3-12 month analyses and the 12-24 month analyses (cf. lines 252-256).

The 12-month recordings were included in both the 3-12 month analyses (as part of the slope calculation), and in the 12-24 month analyses (as the intercept and as part of the slope calculation). Including the 12-month timepoint in both analyses allowed more infants to contribute data to analyses across both developmental windows (since there were only two other visits in the second year). This decision also avoided discontinuity between the developmental windows we examined. That is, only including the 12 month timepoint in the first year analysis would have excluded the period from 12 months to 18 months post-birth from the second year analysis, and only including the 12-month timepoint in the second year analysis would have excluded the period from 9 months to 12 months from the first year analysis.

We have edited the manuscript text to make this allocation of the 12-month timepoint more clear for the reader (text reproduced below).

Lines 316 – 321:

"Individual OLS regressions were run using all available data for each infant across 3 different age ranges: 3-12 months, 12-24 months, and 3 -36 months of age. The 12-month time point was included in both the 3-12 and 12-24 month windows to maximize the sample size of infants contributing to both sets of analyses, and to facilitate more complete coverage of the first and second postnatal years in each respective analysis."

7. lines 178-181: How does BEAPP software "ensure uniform analysis on the EEG data occurred regardless of when the infant was tested and what study group the infant was assigned?"

Again, we apologize for lack of clarity. We sought to emphasize that with BEAPP, all of the data were processed at the same time with the same processing parameters, and with the same empirical criteria for inclusion/exclusion of EEG segments and artifact classification. Thus, there were no differences in the data introduced by variance between manual EEG editors' decisions, or drift in artifact

removal decisions by the same manual editor over the course of preprocessing the data. We have rephrased this point to clarify, as below.

Lines 206 – 209:

“All files were batch processed using the Batch Electroencephalography Automated Processing Platform (BEAPP) software, allowing for the same empirical criteria for artifact removal to be applied to each file in the same way (25).”

8. line 191: “was” → “were”

We have edited the line to make this correction.

Line 226:

“The EEG data were then re-referenced...”

9. lines 255-264: I found this section confusing because (as noted before) it isn't clear how much data were contributed by each infant; moreover I couldn't follow the discussion of the 6-month intercept (or put another way, how 3-month-olds' data were handled).

We agree that the description of how data were handled during the generation of the EEG power parameters could be better clarified. We chose the estimated power at 6 months of age rather than 3 months of age for the initial value so we could more easily compare trends at 6-months of age to prior studies whose earliest timepoint was 6 months of age, and because more infants contributed EEG data at the 6-month timepoint than at the 3-month timepoint (which was added later in the study). The estimated value at 6-months of age was used as the intercept for all infants in the 3-12 and 3-36 month analyses, even if those infants had 3-month data as well as 6-month data, so that everyone would have the same age at intercept. Infants with 3-month old data had that data included when running their OLS regressions, so the 3-month data still contributed to fitting the regression line for those infants (although note that because these were linearized regressions (using $\log(\text{power})$ and $\log(\text{age})$), the 3-month timepoint's data did not affect the regression fit any differently than data from a later timepoint since we were fitting a straight line through all timepoints). We have extensively edited this portion of the text to clarify how and why these parameters were generated. We have also included a conceptual figure illustrating how these parameters were generated for further clarification.

Lines 300 – 336:

“To reduce the number of EEG parameters tested in the primary statistical analyses differentiating outcome and risk groups, the trajectories of EEG summed power within each frequency band across development were summarized by two parameters per frequency band: EEG summed power intercepts and the slope of EEG summed power across age (developmental slope; Figure 1). To generate these parameters, we ran ordinary least squares (OLS) regressions for each infant, modeling that infant's age-related change in EEG summed power within each frequency band. The OLS regressions modeled log-transformed EEG summed power as a function of log

visit age, given the nonlinear, logarithmic age-related changes in log-transformed EEG summed power observed across outcome groups and frequency bands. Modeling log-EEG summed power as a function of log-age linearized the relation between EEG and age (i.e. a linear regression line could be fit between these log-transformed variables). Infants therefore needed EEG recordings from only two visits to calculate this linear regression line, and contribute data to between-participant analyses.

Individual OLS regressions were run using all available data for each infant across 3 different age ranges: 3-12 months, 12-24 months, and 3 -36 months of age. The 12-month time point was included in both the 3-12 and 12-24 month windows to maximize the sample size of infants contributing to both sets of analyses, and to facilitate more complete coverage of the first and second postnatal years in the analyses.

For each developmental window, participants' OLS regression equations were used to generate their individual summed power intercept and linear developmental slope values in each frequency band for between-participant analyses. For the summed power intercept values in the 3-12 month and 3-36 month analyses, the estimated summed power at 6-months of age was calculated from the fitted regression equation for each infant (including infants who had 3-month data). We chose the estimated summed power at 6 months of age rather than 3 months of age for the initial value so we could more easily compare trends at 6-months of age to prior studies whose earliest timepoint was 6 months of age, and because more infants across all risk and outcome groups contributed EEG data at the 6-month timepoint than at the 3-month timepoint (which was added later in the study). In the 12-24 month EEG analyses, 12 months was used for the EEG summed power intercepts for all infants. EEG summed power developmental slopes were calculated as the linear slope parameter estimates from the OLS regression model over each developmental window."

Conceptual Figure (reproduced here in smaller size):

Figure 1: Analysis schematic

Conceptual diagram illustrating how longitudinal EEG parameters were generated and analyzed. **A (for each participant):** at every age, the total, summed power in each canonical frequency band (delta, theta, low alpha, high alpha, beta, gamma) was calculated as the area under the curve of the EEG power distribution (left panel). Growth trajectories of summed power in each frequency band were generated across 1) 3-12, 2) 12-24, and 3) 3-36 postnatal months of age (beta frequency band 3 – 36 month trajectories plotted here, middle panel). Growth trajectories of summed power from each frequency band were linearized by modeling log (summed power) as a function of log (age) for each frequency band; this allowed for the calculation of an estimated intercept (here, at 6 months) and a linear developmental slope for each individual with at least 2 EEG recordings to be submitted to group-level analysis (right panel, here for beta frequency band). **B (group-level effects):** Three types of effects were tested for in the data-driven model construction to differentiate all pairs of groups (here, the low-risk control (LRC) vs. Autism (ASD) group comparison is shown): main effects of differences in intercepts (left panel), and developmental slopes (middle panel), and interaction effects between intercept and slope (right panel). The interaction effect tested whether the relation between intercept and slope varied between groups (e.g. here, individuals in the ASD group have the same slope regardless of having low or high intercept values, but individuals in the LRC group have steeper slope values with higher intercept values).

10. lines 320-361: Do these results concern total EEG power, i.e. collapsed across all frequency bands?

We regret that the results section was not more clear about what was included in these models. The results do not refer to total EEG power, but instead the discrimination performance for the models that included EEG parameters from different frequency bands (which frequency bands and which parameters were included were determined during the data-driven construction of the models). Once the best-fitting models were selected for each developmental window and each group comparison, we generated the model performance statistics summarized in this results section in order to compare how well those best-fit models did at different ages and across different risk/outcome group comparisons. We have edited the results section text and added an initial paragraph describing what the model performance measures reflect and how this step fits with the next section examining the parameters within each model.

Lines 400 – 423:

“Results

To determine when across the first three years after birth EEG power measures best differentiate ASD risk and outcome groups within this sample, a series of logistic regression models were constructed. These logistic regression models differentiated groups (ASD vs. HRA-, ASD vs. LRC, and HRA- vs. LRC) using longitudinal EEG parameters from three developmental windows: 3-12 months, 12-24 months, and 3-36 months [Figure 3, Table 2]. For each group comparison in each developmental window, the EEG power intercepts and developmental slope variables for each frequency band were potential model parameters. Data-driven model construction selected the set of EEG power parameters that best differentiated each pair of groups over each developmental window. Parental education and participant sex were included as covariates for each analysis. First, model performance was compared across the different developmental windows to identify the time period that best differentiated risk and outcome groups. The models constructed with frontal EEG were also compared to models using different spatial layouts (whole-head coverage and temporal-parietal channels; Table 2, Supplemental Results). Finally, the selected EEG power parameters and their significance within each model were compared to identify candidate longitudinal power biomarkers across the different developmental windows.

ASD risk and outcome discrimination across developmental timescales

We first compared how well the different data-driven models performed at differentiating groups across developmental windows by assessing Receiver Operating Characteristic (ROC) curves for the models.”

Lines 460 – 466:

“Next, to assess which frontal EEG power measures differentiated these ASD groups across developmental timescales as potential biomarkers, the EEG power parameters that the data-driven model construction process had selected were examined for each group comparison across each developmental window (Tables 3 – 5; whole-head and temporal-parietal model parameters provided in Supplemental

Tables 1- 6). EEG power for a given frequency band in these models reflects the summed power across all frequencies within that frequency band (summed power)."

11. lines 369-370: What about parents' education? Is that important?

We thank the reviewer for pointing out this omission in the text. The effect of parental education in models was less consistent than participant sex, but was not generally a robust contributor to the models (in almost all cases where it was included, only one of the education parameters was significant, distinguishing parents with 4 year college degrees from those with further education beyond 4 year degrees). We have added text to the results section summarizing this trend across analyses.

Lines 468 – 475:

"With regard to the participant sex and parental education covariates modeled, participant sex was not a significant measure in any model at any age. The parental education covariate was included in many models differentiating the LRC infants from others, but in the majority of comparisons, only the parameter differentiating parents with 4-year college degrees from those with further education was significant (education parameter 2). Supplemental analyses without the sex and education covariates returned similar results to the analyses that included these covariates."

12. Table 1: Earlier (line 229), it is noted that EEG samples must be 40 s to be included in the data set. The means for these values in Table 1 for each group are considerably longer than this, with large SDs. I wondered if the ranges should be included here also, and I was concerned about differences in data quality between short and long samples. Also, what is "Mean retained artifact probability?" (and median)

We agree with the reviewer that further information about the EEG recording lengths would be useful to readers. We have taken several steps to address this concern. We have included the ranges of acquired and usable EEG data in Table 1 as suggested. We have also edited the text to note that neither the lengths of raw or processed EEG data significantly differed between any pair of participant groups (all $p > 0.10$). Thus, the analyses distinguishing these groups from EEG parameters were unlikely to be directly affected by differences in recording lengths between infants or at different timepoints. Baseline EEG power has previously demonstrated high test-retest reliability (alpha estimates of 0.90 and higher) across multiple time scales (within session, across days) in both typically-developing children and clinical populations, including ASD (e.g. Chan et al, 2006; Salinsky et al., 1991; Lund et al., 1995; Fein et al., 1983; Towers et al., 2009). Prior studies have shown that 40 – 60 seconds of clean EEG data is sufficient to achieve such high levels of reliability in absolute power estimates across the scalp (Salinsky et al., 1991; Lund et al., 1995). There are not yet field standards for how many samples of power produce reliable averaged power estimates in infancy (and we have added

text to the discussion section addressing this need). Thus, we followed recent guidelines set out in Cuevas et al., (2014) and by these test-retest reliability studies in children, setting our criteria so that each processed EEG must have at least 40 seconds of clean data to be included in analysis.

To further address the concern about EEG recording length and data quality, we ran supplemental correlation analyses to test whether the length of raw or processed EEG was associated with the amount of artifact in the data, indexed by the percentage of independent components rejected as artifact, and the percent of variance in the EEG retained after artifact rejection. Neither an individual's average raw nor processed EEG file length significantly correlated with any of the artifact rejection variables (all $p > 0.10$) across the EEG data in the 3-12 month, 12-24 month, and 3-36 month analyses. We have edited the methods section text to add this information and to clarify what that various HAPPE-software related measures of data quality reflect (text reproduced below).

Table 1 edited:

Included EEG HAPPE metrics (mean [SD] range)	LRC	HRA-	ASD
Length of raw EEG (seconds)	169.8 (94.3) 44-784	182.9 (128) 44-1067	191.1 (105) 64-594
Good channels (%)	91.9 (4.5) 82.1-100	92.9 (4.5) 82.1-100	91.9 (4.3) 82.1-100
Rejected components (%)	41.3 (11.8) 0-68.8	42.0 (12.7) 5.1-74.3	38.9 (15.0) 0-72.2
EEG variance retained (%)	63.6 (14.9) 32.1-100	63.7 (15.3) 33.6-98.1	68.3 (14.5) 39-100
Mean retained artifact probability	0.16 (0.05) 0.03-0.30	0.17 (0.04) 0.047-0.28	0.16 (0.05) 0.047-0.25
Median retained artifact probability	0.13 (0.08) 0.01-0.34	0.13 (0.06) 0.01-0.31	0.12 (0.06) 0.01-0.25
EEG segments retained (n)	74.6 (40.7) 21-334	81.0 (54.7) 21-474	84.3 (46.5) 31-260

Methods section edits (Lines 240 – 263):

“EEGs were rejected if they had fewer than 20 post-processed good segments (40 total seconds of EEG) in keeping with recommendations for achieving highly reliable EEG power estimates (32, 33), or were more than 3 standard deviations (SD) from the mean on the following HAPPE metrics: percent good channels (3 SD: < 82%), mean retained artifact probability (3 SD: > 0.3; the estimated probability that retained independent components contain any artifact, calculated as an intermediate processing metric before further segment rejection based on remaining artifact), median retained artifact probability (3 SD: > 0.35; the median value for the same artifact probability metric as the mean values), percent of independent components rejected as artifact (3 SD: >84%), and percent of EEG signal variance retained after artifact removal (3 SD: <32%). EEGs with a mean power greater or less than two SD from their outcome group for each frequency band were visually reviewed blind to outcome group status. This led to the rejection of 25 additional EEG recordings (9 LRC EEGs, 11 HRA- EEGs, and 5 ASD EEGs) out of 760 recordings (3% of recordings) and only 1 participant to be dropped from analysis. HAPPE quality metrics and visual inspection rejection rates did not significantly differ between any pair of outcome

groups (all $p > 0.1$, Table 1). Supplemental Pearson correlation analyses assessed whether the lengths of either the raw or post-HAPPE processed EEG files were associated with the amount of artifact in the data (measured directly by the percent of independent components rejected and the percent of variance retained after artifact removal). There were no significant associations between the average lengths of the raw or processed EEG per participant and the amount of artifact detected in the EEG data across the 3-12 month, 12-24 month, or 3-36 month analyses (all $p > 0.10$.)”

Discussion section edits (Lines 648 – 664):

“It is important to acknowledge several limitations of the current study within the context of biomarker development and validation. Here we have identified candidate EEG power biomarkers across multiple developmental windows and spatial configurations, so further research is now required to evaluate the reliability, disorder specificity, and external validity of these measures. We chose to examine baseline EEG power measures as candidate biomarkers in part because baseline EEG power has previously demonstrated high test-retest reliability (Cronbach’s alpha estimates of 0.90 and higher) across multiple time scales (within session, across days) in both typically-developing children and clinical populations, including ASD (e.g. 32, 33, 60–62). Prior studies have shown that 40 – 60 seconds of clean EEG data is sufficient to achieve such high levels of reliability in absolute power estimates across the scalp (32, 33). Therefore, in the present study, we required infants to contribute at least 40 seconds of clean EEG data for analyses to increase the reliability of our model estimates. Future research conducting test-retest analyses of EEG power measures within-participants across consecutive days in infancy is needed to confirm that the high reliability of these EEG power measures extends to the first postnatal months of age.”

- Chan, A. S. & Leung, W. W. M. Differentiating Autistic Children with Quantitative Encephalography: A 3-Month Longitudinal Study. *J. Child Neurol.* **21**, 391–399 (2006).
- Salinsky, M. C., Oken, B. S. & Morehead, L. Test-retest reliability in EEG frequency analysis. *Electroencephalogr. Clin. Neurophysiol.* **79**, 382–392 (1991).
- Lund, T. R., Sponheim, S. R., Iacono, W. G. & Clementz, B. A. Internal consistency reliability of resting EEG power spectra in schizophrenic and normal subjects. *Psychophysiology* **32**, 66–71 (1995).
- Fein, G. *et al.* EEG power spectra in normal and dyslexic children. I. Reliability during passive conditions. *Electroencephalogr. Clin. Neurophysiol.* **55**, 399–405 (1983).
- Towers, D. N. & Allen, J. J. B. A better estimate of the internal consistency reliability of frontal EEG asymmetry scores. *Psychophysiology* **46**, 132–142 (2009).

13. Figure 1: I struggled with Figure 1. It wasn’t clear to me how to relate what is plotted here with the rest of the results. The most salient feature of the three plots (I thought) is the low power for all frequency bands in the HRA- and ASD groups at the 3-month time point. However, it is mentioned (lines 257-258) that not all infants contributed 3-month data, and the number of infants is not provided—perhaps this jump is an artifact? Does it mean something? The other slopes are hard to interpret and there’s not a clear pattern other than obvious differences in power between different bands in all three groups. Nevertheless, the 3-12 month slope in delta and theta bands

(Table 3) assume importance in the overall interpretation of the study. Thus it may be important to describe what data are included in the analyses (echoing previous comments).

We agree that Figure 1 (now Figure 2) should be re-organized to help detect the differences picked up in the statistical analyses. We have reformatted the figure so that each frequency band is plotted with different lines for each group to facilitate these comparisons.

To the reviewer's point about the jump in power from 3 to 6 months of age, currently not many papers have looked at EEG power across the first year that include ages as low as 3 months, but those that have examined frontal power within the same age range find highly-similar patterns to ours (Cornelissen et al., 2015; Jing et al., 2010). That is, there is much lower power at 3 months of age across the power spectrum compared to the second half of the first postnatal year, and the changes in EEG power with age seem to be highly nonlinear and logarithmic for all frequencies (see figures below). Thus, given the limited current knowledge of these trajectories, we do not believe the trends in this dataset are suspect, but we hope further studies explore EEG power changes at these very early ages to better understand this pattern.

From Cornelissen et al. 2015 (showing much lower power in 0-3 mo olds compared to 4-6 mo olds):

From Jing et al. 2010 (BF, MF, SF are all different groups of infants):
Frontal 3-6 Hz power:

9-12 Hz power:

12-30hz power:

Cornelissen, L., Kim, S.-E., Purdon, P. L., Brown, E. N. & Berde, C. B. Age-dependent electroencephalogram (EEG) patterns during sevoflurane general anesthesia in infants. *Elife* **4**, (2015).

Jing, H., Gilchrist, J. M., Badger, T. M. & Pivik, R. T. A longitudinal study of differences in electroencephalographic activity among breastfed, milk formula-fed, and soy formula-fed infants during the first year of life. *Early Hum. Dev.* **86**, 119–125 (2010).

Revised Figure (formerly Figure 1, now Figure 2):

Figure 2: EEG power from 3 – 36 months of age

Mean log of EEG power, calculated as the sum of the power across each frequency band, is shown for each frequency band over all visit ages for each outcome group, low-risk control (LRC), high-risk without Autism (HRA-), and high-risk with Autism (ASD) for the frontal region of interest. Lines connecting power values across visit ages are to aid visualization. Error bars are 95% confidence intervals around the mean.

Reviewer #2 (Remarks to the Author):

14. My main concern is with the use of the term “biomarker.” Increasingly, studies label any physiological measures sensitive to group differences as biomarkers without explicitly demonstrating that they indeed meet the criteria. The retrospective design is helpful as the first step for identifying candidate biomarkers for further testing. However, and the authors acknowledge this in the limitations, they need to be validated in new samples and ideally, by other, independent teams. In the current study, 18 different frontal EEG power measures (6 frequency bands x 3 metric types) were sensitive to some diagnostic group differences in at least one of the reported analyses. Which of these measures (or a combination of them) would the authors recommend for further, more detailed validation?

We agree with the reviewer that it would be premature to call the EEG power parameters biomarkers at this stage of analysis. Therefore, we have toned down the language throughout the text to refer to the EEG power parameters that distinguish groups as candidate biomarkers in the revised text.

As the reviewer notes, our candidate biomarkers included parameters from multiple EEG bandwidths. As we tested for the combination of EEG power variables that together could distinguish groups, the set of power parameters included in each data-driven model constitutes the candidate biomarker for that developmental window that should be subjected to further validation. Almost all models included power parameters from multiple bandwidths, suggesting the pattern of power across bandwidths facilitates better differentiation of risk and outcome groups than parameters in any single bandwidth. We have added text to the discussion section to clarify this point and put these multivariate candidate biomarkers into context (reproduced below).

In addition to the sets of parameters that together best differentiated groups, we also note individual physiological differences that were consistently selected across models as significant parameters that may shed light on physiological mechanisms altered in ASD (e.g. the delta and theta slopes, gamma parameters). We recommend further investigation of the processes giving rise to these differences to better understand the pathophysiology of ASD (although these parameters alone would not better differentiate groups).

In response to the reviewer's thoughtful points about biomarkers we have also added several paragraphs to the discussion section that situate the present findings within this larger context of biomarker development and elaborates the next steps to assess whether these candidate parameters predict group status at the individual level (reproduced below).

Lines 598 – 617:

“Importantly, we identified the EEG power pathophysiology during each developmental window that best differentiates ASD risk and subsequent ASD outcomes in our sample. As we tested for the combination of EEG power variables that together could distinguish groups, the set of power parameters included in each data-driven model constitutes the candidate biomarker for that developmental window. Almost all models included power parameters from multiple frequency bands, suggesting the pattern of power across bands facilitates better differentiation of risk and outcome groups than parameters in any single frequency band. These results are consistent with Wang and colleagues’ synthesis across studies of baseline EEG power suggesting that an atypical distribution of power across the spectrum distinguishes individuals with ASD from neurotypical individuals (46). Accordingly, recent interventions seeking to modulate frontal EEG power in those with ASD have assessed patterns of change across the power spectrum as the biological target (47). Evaluating multiple EEG frequency bands simultaneously may thus facilitate both efforts to differentiate outcomes and to leverage EEG power in interventions at the physiological level.

Across models, we also identified trends for specific EEG frequency bands that significantly contributed to group differentiation and may reflect physiological mechanisms altered in ASD development. We found that developmental changes in low-frequency power stratified both ASD risk status and diagnostic outcome...”

New limitations/next steps section of the discussion (lines 648 – 698):

“It is important to acknowledge several limitations of the current study within the context of biomarker development and validation. Here we have identified candidate EEG power biomarkers across multiple developmental windows and spatial configurations, so further research is now required to evaluate the reliability, disorder specificity, and external validity of these measures. We chose to examine baseline EEG power measures as candidate biomarkers in part because baseline EEG power has previously demonstrated high test-retest reliability (Cronbach’s alpha estimates of 0.90 and higher) across multiple time scales (within session, across days) in both typically-developing children and clinical populations, including ASD (e.g. 32, 33, 60–62). Prior studies have shown that 40 – 60 seconds of clean EEG data is sufficient to achieve such high levels of reliability in absolute power estimates across the scalp (32, 33). Therefore, in the present study, we required infants to contribute at least 40 seconds of clean EEG data for analyses to increase the reliability of our model estimates. Future research conducting test-retest analyses of EEG power measures within-participants across consecutive days in infancy is needed to confirm that the high reliability of these EEG power measures extends to the first postnatal months of age.

Second, further testing across clinical populations is required to determine the specificity of these EEG power candidate biomarkers. The present study could not disentangle whether the EEG power parameters differentiating the infants with ASD were specific to the familial-risk group or would generalize to other ASD-risk groups. Additionally, the candidate biomarkers we identified may reflect disruptions in processes affected in other neurodevelopmental disorders as well (e.g. language acquisition, 11). However, infants with high-familial risk who are not diagnosed with ASD frequently manifest subclinical and clinical disruptions in the same domains as the high-risk infants with ASD, and develop other disorders at elevated rates (43, 63–67). The candidate biomarkers we identified distinguishing the high-risk infants with and without ASD achieved very high levels of accuracy, though, suggesting some specificity for ASD diagnosis in the present study. Additional research across ASD-risk and neurodevelopmental disorder-risk populations is needed to confirm the diagnostic specificity of these EEG power measures before they may be used as diagnostic biomarkers.

Lastly, further research is required to assess the external validity of the candidate biomarkers identified in the present analyses. While this study had a greater number of infant siblings with ASD diagnoses relative to prior reports, the present sample is still limited, so these findings should be replicated with larger samples in future research. We were well powered to achieve the study’s primary goal of evaluating model performance within our longitudinal sample across different developmental windows, but our sample size prevented us from validating model performance in out-of-sample sets (e.g. split-half training and testing sets).

Additionally, the study’s over-recruitment of high-risk infants is a strength, but the ratio of high-risk to low-risk infants is therefore not representative of the general population. So, the positive and negative predictive values for models comparing these groups may be different in the general population. Future research testing the external validity of these power measures should also examine how robust the patterns identified here are across EEG acquisition equipment and parameters. To facilitate this

type of future assessment, here we used a brief, naturalistic baseline paradigm, as collecting several minutes of EEG at baseline may prove more scalable than collecting event-related EEG, and we used standardized, open-source processing software that we developed for use across various EEG systems and files with different acquisition parameters.”

15. To be clinically meaningful, a biomarker needs to be specific to the diagnostic condition. In the current paper, the multiple bands that differentiated ASD and/or high-risk from low-risk infants would reflect a variety of neural processes (the authors noted inhibition and experience-dependent plasticity as some of the possibilities). Could the authors comment on how one would determine whether the observed resting EEG power differences are specific to ASD or reflect more general language difficulties, executive function deficits, etc.?

We share the reviewer’s interest in discerning which processes these candidate biomarkers reflect. Notably, the high-risk infants who do not go on to have ASD do frequently demonstrate subclinical difficulties in the same domains affected in ASD (e.g. language difficulties, executive function deficits), or even receive diagnoses for other neurodevelopmental disorders. Therefore, the EEG power parameters distinguishing the high-risk ASD infants from the high-risk infants without ASD in these analyses reflect some specificity for ASD.

To fully address this question of marker specificity for ASD, multiple longitudinal datasets following different familial risk populations (e.g. specific language impairment, ADHD, anxiety, dyslexia, etc) would need to be combined, a significant but worthwhile endeavor for future research. We have added this consideration to the limitations section of the discussion section (See paragraphs above for comment #14).

16. On a related note, the manuscript suggests that frontal EEG power is the most informative marker, but the rationale for initially selecting this location was not clearly established.

We agree with the reviewer that more justification for selecting the frontal ROI would be helpful. Please see comment #3 for our edits to the introduction to make this justification more clear to the reader. However, to enable further assessment of this a priori frontal ROI’s discrimination performance, we have now also included the full set of 10-20 models for each developmental window, and models based on temporal-parietal EEG power in keeping with your suggestion below.

17. Furthermore, the empirical evaluation of spatial specificity might have been confounded by differences in signal quality. The frontal EEG power metrics came from a large cluster of spatially contiguous electrodes with highly correlated data, while the whole-head dataset was an average of spatially distributed single channels. Thus, the frontal EEG likely had better

signal-to-noise ratio, which could explain greater sensitivity to the diagnostic group differences compared to the whole-head average. The goal of evaluating spatial specificity of the frontal EEG biomarkers could have been tested more directly if the frontal cluster data were compared to similar-sized clusters in temporal, parietal, or occipital regions.

We appreciate this point about signal quality. Though our primary goal was to identify the developmental window when frontal EEG power best discriminated outcome groups, we then planned to examine the 10-20 configuration of electrodes to see whether the frontal findings were specific to frontal electrodes or generalized across the scalp. For this comparison, we were examining exactly this point that you raise, whether the distributed channels in the 10-20 layout were sufficient to differentiate groups, or whether non-standard, spatially-clustered electrode configurations (not necessarily frontal) may be needed. We have edited the discussion text to include this point explicitly.

Although we did not aim to determine if frontal was the best performing ROI of any potential ROI in this manuscript, we agree that assessing another spatially-localized cluster may provide important context for these results. We have therefore calculated power over a bilateral temporal-parietal cortical ROI (with similar dense spatial coverage as the frontal ROI) for the same participants and performed the full set of analyses included in the original manuscript. We have included these analyses and results for all three developmental windows as supplemental materials.

Of note, frontal EEG power in the first year postnatally still provides the most accurate differentiation of ASD diagnostic outcomes. Both spatially-localized ROIs outperformed the whole-head models in the first postnatal year. The frontal ROI outperformed the temporal ROI over the 3 year period as well, and slightly bested the whole-head model over the 3 years (though the whole-head model outperformed the spatially-localized models when considering the 2nd postnatal year only). Thus, it seems important to consider both degree of spatial localization of electrodes and where that spatial localization occurs, especially for discriminating outcomes over the first postnatal year. We have added text to the discussion section to touch on these points.

Discussion text (lines 552 – 569):

“Our findings also indicate that the spatial localization of EEG power measurements matters in distinguishing outcomes. The a priori clustered frontal EEG region of interest and a clustered temporal-parietal layout examined in supplemental analyses both provided better differentiation than the averaged 10-20 standard layout in the first postnatal year, suggesting sparse electrode configurations like those used in clinical settings currently may not provide the optimal layout for measuring early EEG biomarkers of ASD (see Supplemental Material). The more densely clustered EEG layouts may have benefited from a higher signal-to-noise ratio than the whole-head layout. Notably, the frontal region of interest also provided better differentiation in the first year than the clustered temporal-parietal region, suggesting the spatial specificity of the EEG power measures is important for discriminating groups. At later developmental windows, the densely clustered and whole-head configurations offered

different strengths, such that the frontal and temporal-parietal layouts largely achieved higher sensitivity rates, while the whole-head layout had largely higher specificity rates. Thus, both the spatial location and the spatial density of EEG channels are important factors to consider in EEG-derived biomarker development to differentiate ASD outcomes, especially early in postnatal development.”

Temporal ROI model performance (now included in Table 2):

Models	Discrimination Rate (CI ₉₅)	Sensitivity [†]	Specificity [†]	PPV	NPV
Temporal 3 -12 months					
ASD vs. HRA-	83.4% (72.9 – 93.9%)	72.73	86.27	69.57	88
ASD vs. LRC	82.2% (68.7 – 95.8%)	64.71	96.49	84.62	90.16
HRA- vs. LRC	77.4% (68.6 – 86.3%)	64.58	78.95	72.09	72.58
Temporal 12 - 24 months					
ASD vs. HRA-	76.9% (66.2 – 87.6%)	80	65.31	54.05	86.49
ASD vs. LRC	89.4% (81.3 – 97.4%)	90	75	60	94.74
HRA- vs. LRC	80.9% (72.6 – 89.2%)	83.67	68.52	70.69	82.22
Temporal 3 - 36 months					
ASD vs. HRA-	83.6% (75.1 – 92.2%)	67.74	85.92	67.74	85.92
ASD vs. LRC	87.0% (77.5 – 96.6%)	84	83.87	67.74	92.86
HRA- vs. LRC	75.2% (66.8 – 83.7%)	72.31	72.58	73.44	71.43

**Tables with the model parameters for each developmental window also included as Supplemental Tables 4-6.*

18. A biomarker, especially a diagnostic one, also needs to be reproducible: the current study did not include a test of reliability of the EEG power measures. Different EEG frequency bands and power metrics (intercept or slope) at different ages were predictive of the later diagnostic outcomes, with some losing their predictive value over time. The authors attributed this potential instability to extensive neural development in the first three years of life, but it is unclear whether the underlying pathophysiology of ASD reflected by the individual frequency bands would have changed significantly from 6 to 12 to 24 months, etc.

We agree with the reviewer that demonstrating reproducibility is an important step in evaluating biomarkers, and this issue informed several of our decisions in the current study (see text added to the discussion section to address this point, provided in response to comment #14 above). To our knowledge, there is no formal assessment of test-retest reliability in infancy for EEG power, which would require assessing the same infants over consecutive days to avoid the confound of rapid development co-occurring during this period (this would be a worthwhile, if challenging, endeavor for a future study, and see Van der Velde et al.,

2019 showing high test-retest reliability of EEG-derived network connectivity measures in infancy).

We did find that some EEG power metrics lost their *relative* predictive value over time. We should have stated this more clearly in the discussion section (and we have edited the text accordingly, reproduced below). Our data-driven modeling was executed to select the most parsimonious set of parameters that achieved the best differentiation. Thus, the differences in EEG power measures found in the 3-12 month model may still exist in later developmental windows, but they may not be the most robust differences anymore by three years of age as other developmental dynamics have emerged. Consistent with this account, 21 of 27 frontal EEG parameters in the 3-12 month model were also included in the 3-36 month model (and 29/35 total model parameters were included in the 3-36 month model). The general difference between these two developmental windows was the inclusion of higher-frequency parameters in the 3-36 month model, not the omission of lower-frequency differences observed earlier (although they differentiate ASD outcomes at lower rates over the 3-36 month window than in isolation during the first year). The inclusion of higher frequency parameters over later developmental periods is consistent with the observed developmental trend that higher-frequency dynamics emerge later than lower-frequency dynamics. So, while the early-emerging pathophysiology of ASD may remain stable across ages, the developmental emergence of increasingly complex dynamics reflected across more bandwidths may change which differences together best distinguish ASD risk or outcome groups in our analyses. Moreover, new adaptive and compensatory changes, also reflected in the EEG power spectrum, are likely to occur over time in response to this underlying pathophysiology.

Discussion edits (lines 570 – 586):

“Our data-driven modeling approach to identify pathophysiology highlights the importance of characterizing the longitudinal development of candidate biomarkers in several ways. First, this approach selected the most parsimonious set of parameters that differentiated groups within each developmental window, allowing us to examine whether the most robust EEG power discriminators changed as a function of age. For example, differences in EEG power observed in the first year may persist but may no longer be the most robust discriminators in later developmental windows as other developmental dynamics emerge, including adaptive and compensatory changes. Indeed, across timescales from 12 months to 3 years, we observed developmental changes in which parameters provided the best differentiation. Though early-emerging differences largely remained robust discriminators at later ages, we also noted a general shift from significant low frequency predictors across the first year towards additional significant higher frequency predictors across the second and third years. This delay in high-frequency candidate biomarkers showing robust differentiation recapitulates maturation patterns of EEG power spectra observed in prior studies during this same period (44).”

Chan, A. S. & Leung, W. W. M. Differentiating Autistic Children with Quantitative Encephalography: A 3-Month Longitudinal Study. *J. Child Neurol.* **21**, 391–399 (2006).

- Salinsky, M. C., Oken, B. S. & Morehead, L. Test-retest reliability in EEG frequency analysis. *Electroencephalogr. Clin. Neurophysiol.* **79**, 382–392 (1991).
- Lund, T. R., Sponheim, S. R., Iacono, W. G. & Clementz, B. A. Internal consistency reliability of resting EEG power spectra in schizophrenic and normal subjects. *Psychophysiology* **32**, 66–71 (1995).
- Fein, G. *et al.* EEG power spectra in normal and dyslexic children. I. Reliability during passive conditions. *Electroencephalogr. Clin. Neurophysiol.* **55**, 399–405 (1983).
- Towers, D. N. & Allen, J. J. B. A better estimate of the internal consistency reliability of frontal EEG asymmetry scores. *Psychophysiology* **46**, 132–142 (2009).
- Van der Velde B, Haartsen R, Kemner C. Test-retest reliability of EEG network characteristics in infants. *Brain Behav.* 2019;9(5):e01269.

Other comments:

19. The frontal electrodes used in the analysis appear to represent two lateralized clusters with a narrow midline “connector” (also, Supplementary Figure 1 may have an incorrect color code for channel 3 in the 64-channel net). Previously, this group (Gabard-Durnam et al., 2015) suggested that hemisphere asymmetry in frontal EEG may reflect an autism endophenotype in infants. Thus, it would be interesting to know why hemisphere differences were not considered in the present study.

We agree with the reviewer that hemisphere asymmetry in frontal EEG power would be an interesting set of measures to consider moving forward. We did not include those measures here in part because prior studies had shown there were group differences in frontal power across the bandwidth spectrum, while we had only previously looked at asymmetry in the alpha band (Tierney et al. 2012 did not find significant differences between groups in power asymmetry, although we collapsed across ages for those post hoc tests). For this initial assessment we aimed to determine which frequency band differences may be most meaningful in discriminating ASD groups, and therefore chose to start with nonlateralized power where a greater range of differences had been noted already. We also took into consideration sample size restrictions on the number of potential parameters to test (testing the full set of asymmetry parameters would mean another 18 variables). A future manuscript that runs the same sets of analyses with asymmetry measures to see whether they can outperform the power measures in the current manuscript would be an excellent next step for our group.

20. Previously, Tierney et al. (2012) suggested that age-dependent group differences in EEG power may be less informative than developmental trajectories of change. Using the current large data set, could the authors comment on whether the slope of EEG power is indeed a more robust marker than EEG power at a single age?

The data driven parameter selection we used in this paper did include power slope and interaction terms in every model we tested, suggesting there is unique and highly-useful information to be gained from two timepoints of data compared to 1 timepoint of data. Roughly 2/3 of any given model’s parameters involved the slope

parameters! So, having the intercept and slope values to construct longitudinal trajectories does seem more informative than just having the intercept values.

It is difficult to say whether, in isolation, the trajectory's variables of slope vs. power at a single age is more robust given the analyses we ran, because we did not aim to test EEG power at each single age or slopes in isolation without other model parameters. However, both power intercepts and slope (or interaction) parameters were significant parameters in all models at roughly the same rates, suggesting where infants "start" with EEG power at early ages, as well as how their EEG power changes with age, are important indicators for outcome.

21. The biggest challenge to using EEG as a diagnostic biomarker to date has been the inability to interpret the data at the individual level. This is a major difficulty for the field of human neurophysiology because the observed EEG signal may vary based on equipment, acquisition and signal processing routines that tend to be different across labs. Given the authors' expertise and interest in biomarker development, could they comment on the next steps toward moving from group-level to individual-level results?

Yes, we agree with the reviewer that there are currently challenges in applying physiological biomarkers at the individual level for exactly these kinds of reasons. We are actively working on some of these issues across projects, like the need for standardized signal processing routines that can align EEG data from different systems or with varied acquisition settings. For the present manuscript, we processed everything through our standardized software packages (HAPPE and BEAPP), which we designed to address these needs. We have also made the software open-source and freely available to provide these tools for the field to move towards standard processing routines. We have also focused heavily on baseline EEG parameters (like in the current manuscript) because the acquisition conditions are easy to reproduce, the collection time is fairly brief, and this paradigm may one day translate into a clinical setting to interpret data at the individual level. It will be important in future work, most likely through collaborative efforts across labs, to join datasets from different equipment and locations to examine which candidate measures translate across these systems most robustly (or we may find we need to standardize equipment for the purpose of making individual-level predictions). This would also provide large enough sample sizes of infants who go on to have ASD to robustly test individual-level prediction. But first, there is still work to be done in figuring out which physiological measures distinguish outcomes at the group-level at different ages as candidates for that further analysis (which we are actively working on across ongoing studies).

Reviewer #3 (Remarks to the Author):

22. Please list maximum number of channels rejected with BEAPP (this is not automatically regulated by beapp as far as I know so can be a concern). Was HAPPE done on all 64/128 channels or just the ones used in the analyses? If just on the reduced number of electrodes, then rejecting/interpolating more

than a single channel is problematic (usually 5% or fewer interpolated channels is a general guideline), especially given the referencing to average reference.

We agree that we did not describe this processing step in enough detail in the text. The reviewer is correct that BEAPP does not regulate the number of channels rejected, but it reports how many channels were marked good channels during processing. We used this report to reject any EEG file that had fewer than 82% good channels during processing (more than 3 standard deviations away from the complete sample mean). In the included sample, there were no significant differences in the % channels marked good between ASD groups (all $p > 0.10$), and the mean % channels marked good was $\sim 92\%$ (i.e. mean % channels rejected was $\sim 8\%$, or 2-3 channels across net types). This was a skewed distribution, with the majority of the EEGs (62%) having greater than 92% good channels (fewer than 2-3 bad channels). To our knowledge, this is well within the typical range for infant/child studies and studies with the same infant-sibling populations in the field. Interpolation rates between 10-20% are frequent, when a channel limit is reported at all (e.g. Elsabbagh et al., 2015; Orekhova et al., 2014; Xie & Richards, 2017), although some groups interpolate as many as 35% channels (e.g. Kouider et al., 2013). To better inform the reader of our interpolation rates, we have added the mean, standard deviation, and range of % good channels for each group to Table 1 demographics for the reader (excerpt below for reference).

As the reviewer points out, we ran HAPPE on more channels than we used in analyses precisely so we could reasonably employ an average re-reference strategy. However, given the length of the baseline EEG recordings, we could not run all 64 or 128 electrodes without sacrificing artifact removal effectiveness during independent component analysis, so we processed a spatially-distributed subset of channels that provided equivalent spatial coverage between the 64 and 128 channel nets (29 and 39 channels processed respectively). We have edited the manuscript text and Supplemental Figure 1 (with net layouts) to include the channels that we processed in addition to the channels that we analyzed (text edits below for reference).

- Elsabbagh, M., Bruno, R., Wan, M. W., Charman, T., Johnson, M. H., Green, J., & Team, T. B. (2015). Infant Neural Sensitivity to Dynamic Eye Gaze Relates to Quality of Parent–Infant Interaction at 7-Months in Infants at Risk for Autism. *Journal of Autism and Developmental Disorders*, 45(2), 283–291.
- Kouider, S., Stahlhut, C., Gelskov, S. V., Barbosa, L. S., Dutat, M., De Gardelle, V., ... Dehaene-Lambertz, G. (2013). Supplementary Materials for A Neural Marker of Perceptual Consciousness in Infants This PDF file includes Materials and Methods Supplementary Text Figs. S1 to S5 Full References. *Science*, 340, 376.
- Orekhova, E. V., Elsabbagh, M., Jones, E. J., Dawson, G., Charman, T., & Johnson, M. H. (2014). EEG hyper-connectivity in high-risk infants is associated with later autism. *Journal of Neurodevelopmental Disorders*, 6(1), 40.
- Xie, W., & Richards, J. (2017). Development of infant sustained attention and its relation to EEG oscillations: An EEG and cortical source analysis study NeuroDevelopmental MRI Database View project Infant Visual Attention, Perceptual Processing, and Recognition Memory View project.

Table 1 edited:

Included EEG HAPPE metrics (mean [SD] range)	LRC	HRA-	ASD
Length of raw EEG (seconds)	169.8 (94.3) 44-784	182.9 (128) 44-1067	191.1 (105) 64-594
Good channels (%)	91.9 (4.5) 82.1-100	92.9 (4.5) 82.1-100	91.9 (4.3) 82.1-100

Methods section edit (lines 209 – 215):

“To optimize artifact rejection performance given the lengths and sampling rates in the EEG data, a spatially-distributed subset of channels providing whole-head coverage was processed through HAPPE (64-channel net –2, 3, 6, 8, 9, 11, 12, 13, 15, 16, 17, 21, 24, 25, 27, 28, 34, 37, 40, 46, 49, 50, 52, 53, 54, 57, 58, 61, 62; 128-channel net –3, 4, 9, 11, 13, 19, 20, 22, 23, 24, 27, 28, 33, 36, 40, 41, 45, 46, 47, 52, 58, 62, 70, 75, 83, 92, 96, 98, 102, 103, 104, 108, 109, 112, 117, 118, 122, 123, 124; Supplemental Figure 1).”

23. Resampling to 250 Hz and low passing at 100 Hz violates the Nyquist rule (as it is generally applied to EEG) where the anti-aliasing low pass filter should be less than 1/3 of sampling frequency (so low pass should be somewhere around 80 Hz maximum for a 250 Hz sampling rate). This could impact gamma and beta findings if high gamma gets aliased to beta, which could be happening given the large amount of beta that is apparently occurring in this data.

We apologize that our order of processing steps was not more clear in the methods text. We first filtered the data (100hz) and then resampled to 250 Hz to avoid the aliasing problem that the reviewer describes when these steps are reversed (i.e. resampling and then low pass filtering). We have clarified the methods section to state that we filtered first and then resampled the EEG data.

As the reviewer notes, the Nyquist rule applies to which frequencies can be examined with integrity (one cannot faithfully represent frequencies greater than $\frac{1}{2}$ sampling rate). Thus, there will be some differences between ERP analyses and time-frequency analyses because they use filters differently. To our knowledge, the $\frac{1}{3}$ – $\frac{1}{5}$ sampling rate guideline for filtering from Steve Luck comes from the context of ERP analysis, where all frequencies below the filter cutoff are then implicitly examined in the time domain (as an ERP). However, here for our power analysis, we used the low-pass filter as an intermediate step to restrict what frequencies are fed to the independent component analysis, rather than to determine what frequencies we examine in analysis. The 100hz filter we applied is well below the $\frac{1}{2}$ sampling rate specified by the Nyquist rule as used in time-frequency analyses (Cohen, 2014). Here, we only examine frequencies up to 50Hz, which are all below the benchmark of $\frac{1}{3}$ of the sampling frequency.

We thank the reviewer for catching the point of confusion about why it seems we have high beta/gamma power in this sample, and we clarify below in response to comment # 27 (our measure is total power in the beta range, rather

than power density, or power/Hz that is also frequently used in the literature, which does indeed follow the 1/f distribution. We realize this was not explicitly stated in the original manuscript, so we have taken steps to make this point clear).

Cohen Mike. *Analyzing Neural Time Series Data: Theory and Practice*. (MIT Press, 2014).

Methods section edit (lines 215 – 220):

“For each EEG, a 1 Hz digital high-pass filter and a 100 Hz low-pass filter was applied in preparation for independent component analysis (26). Data sampled at 500 Hz were then resampled with interpolation to 250 Hz as recommended for HAPPE processing (resampling was performed after filtering to avoid aliasing higher frequencies when resampling).”

24. Impedances at 100 kohm is pretty high. The conservative standard for EGI is 50 kohm with 75 being a more liberal threshold. 50 kohm should be easily attainable with infant scalp. Is there a reason why impedances were allowed to be so high?

We appreciate the reviewer’s concern about impedance thresholds. EGI does recommend 50kohm thresholds for studies with adults (where there is unlimited time to adjust impedances without participants getting fussy) and those in non-electrically shielded rooms. However, we spoke with their representatives at the start of the study and use 100kohm based on those conversations as a way to save time and initiate testing more quickly (given the limited attention/compliance timeframes with infants) without signal degradation. (Several other infant EEG labs using this same system apply the 100kohm threshold for the same reasons.) The EGI amplifiers used in the study are all high-impedance amplifiers (200Mohm and upwards), as the reviewer notes. These amplifiers are built to facilitate accurate signal collection under conditions of skin-electrode impedance values upwards of 200 kohms without appreciable signal degradation (see Ferree et al., 2001 for the empirical test of these amplifiers’ performance under variable skin-electrode impedances.) Since signal attenuation occurs as a function of skin-electrode/amplifier impedances, 100kohm skin-electrode impedance thresholds are not a problem for EEG signal collection, and in line with recommendations that the skin-electrode impedances be at least 1% of amplifier input impedance for optimal signal recording (Picton et al., 2000). Of note, the 100kohm threshold would increase sensitivity to line noise by a small amount (Ferree et al., 2001), but we performed all testing in electrically-shielded rooms, so our line noise is greatly attenuated relative to other recording setups and this is not a concern (confirmed with EGI as well). We have edited the text to note more clearly that these impedance thresholds were set in the context of electrically-shielded testing to clarify for the reader.

Methods section edit (lines 195 – 197):

“Impedances were kept below 100K Ω in accordance with the impedance capabilities of the high-impedance amplifiers inside the electrically shielded room (23).”

Ferree, T. C., Luu, P., Russell, G. S. & Tucker, D. M. Scalp electrode impedance, infection risk, and EEG data quality. *Clin. Neurophysiol.* **112**, 536–544 (2001).

Picton, T. W. *et al. Guidelines for using human event-related potentials to study cognition: Recording standards and publication criteria.* (2000).

25. The results section states the frequency bands with significant differences between groups, but perhaps a bit more description is warranted here on how they differ (e.g. gamma decreases in one group over time but not another), rather than waiting until the discussion section for this.

We thank the reviewer for this suggestion and we have added more description for each of the parameters in the results section to describe these changes. (In the context of the logistic regressions discriminating group status, the parameters have a different interpretation than in linear regression.)

Example results edit (lines 483 – 493):

“Higher frequency bands, beta and gamma, were only selected in data-driven models differentiating ASD infants from others, such that lower levels of high-frequency power at age 6 months increased the log-odds of the child having ASD over and above the other selected frequency band parameters (e.g. beta power intercept, $p = 0.022$). Delta and theta slopes during this period were significant individual indicators of high-risk for ASD (ASD and HRA-) relative to LRC status, with steeper delta and less-steep theta slopes over the first postnatal year increasing the log-odds of the child belonging to the high-risk groups in the context of the other selected frequency band parameters (ASD: delta slope $p = 0.006$, theta interaction $p = 0.013$; HRA-: delta slope $p = 0.022$, theta slope $p = 0.003$).”

26. Discussion says that frontal electrodes results are better than whole head but the results say findings were largely the same. Please reconcile.

We agree we should clarify this point in the manuscript. In the results section, we meant to write that the bandwidths selected to be in both the frontal and whole-head models were largely the same, although they achieved different levels of success in differentiating groups (e.g. delta, theta, gamma parameters calculated from the frontal region outperformed delta, theta, and gamma parameters from the average whole-head configuration). In light of running supplemental analyses on an additional temporal-parietal region of interest (as requested by reviewer 2), we have edited the discussion text to clarify these points across all of the spatial layouts we tested.

Discussion text edits (lines 551 – 568):

“Our findings also indicate that the spatial localization of EEG power measurements matters in distinguishing outcomes. The a priori clustered frontal EEG region of interest and a clustered temporal-parietal layout examined in supplemental analyses both provided better differentiation than the averaged 10-20 standard layout in the

first postnatal year, suggesting sparse electrode configurations like those used in clinical settings currently may not provide the optimal layout for measuring early EEG biomarkers of ASD (see Supplemental Material). The more densely clustered EEG layouts may have benefited from a higher signal-to-noise ratio than the whole-head layout. Notably, the frontal region of interest also provided better differentiation in the first year than the clustered temporal-parietal region, suggesting the spatial specificity of the EEG power measures is important for discriminating groups. At later developmental windows, the densely clustered and whole-head configurations offered different strengths, such that the frontal and temporal-parietal layouts largely achieved higher sensitivity rates, while the whole-head layout had largely higher specificity rates. Thus, both the spatial location and the spatial density of EEG channels are important factors to consider in EEG-derived biomarker development to differentiate ASD outcomes, especially early in postnatal development.”

27. I have some concerns about the power results in figure 1. Power usually follows a 1/f distribution, with low frequencies having highest power yet the log power in figure 1 shows relatively low power in low frequencies and very high power in beta band, which is not consistent with any other developmental studies that I am aware of. Residual motion artifact might explain the increase in beta band power, or aliasing from gamma. Alternatively several infant EEG papers recently published by some of these coauthors and presumably collected in the same lab set-up show this increase in beta relative to the other frequencies, which may point to something that may not generalize to other research groups. Although the groups did not differ on artifact rejection criteria via HAPPE, it is important to identify the source (theoretical neural or artifactual) for this unusual finding in case different groups have different contributing factors to this beta increase (i.e. one is more prone to certain types of artifact than the other at different developmental windows, contributing to group trajectory differences in specific frequency bands). Do the authors have a mechanistic theory for why the power spectra are shaped this way? Tierney 2012, reporting over similar age range and electrodes, shows a standard 1/f distribution in power, as well as a decrease in beta and gamma power with age that is dissimilar to what is reported here. Given the limited number of studies available for comparison, how might we reconcile these findings?

We recognize that our measure of power was not sufficiently described in the manuscript and we thank the reviewer for bringing this point to our attention. Here, we consider total summed power within each frequency band rather than power density (power/Hz). The measures are directly related, but as the reviewer notes, have very different distribution shapes. Namely, while beta and gamma have very low power/Hz, given the number of frequencies contributing to their power estimates, they each have high summed power across the band. The power density distributions for all of our groups follow general 1/f distributions, with similar distribution shapes at each age across groups (see below for figure). The distinctive alpha peak and some of the 1/f shape emerges over the first year (e.g. Xiao, Shida-

Tokeshi et al., PlosOne 2018), which we see across groups in this study as well. All ASD risk and outcome groups show a beta peak at 3 months that gradually fades by 3 years of age. This beta peak is sometimes seen over frontal electrodes in the quiet, awake, baseline state (e.g. Kirov et al., 2009) and computational modeling suggests it may reflect changes in thalamo-cortical excitatory-inhibitory transmission (e.g. Hashemi et al., 2017, PlosONE modeling this effect under various states of inhibition, which is actively changing during the 3 month – 3 year window in this study). We hope further research is conducted to more thoroughly characterize the expected power distribution shape across early postnatal development, though.

Confusingly, both total, summed power and power/hz are used and reported as “power” in the literature. At the time of analysis, the BEAPP software’s standard outputs were summed power rather than power density, and since the two measures would generate the same results with just different value scales (since power/hz is just summed power/#frequencies within the band), we did not transform summed power into power density for the manuscript’s analyses.

We have taken several steps to clarify our measure of power in the text and figures to avoid future confusion. 1) We have edited the methods section to clearly describe how power was derived for these analyses. 2) We have changed the text to refer to “summed power within x-band” rather than “x-band power” throughout the text. 3) We have included a conceptual figure (new Figure 1) illustrating our power measures in the context of power distributions. 4) We have added a supplemental figure illustrating the power distributions for each age and each group so the reader can confirm that these distributions follow the general expected 1/f pattern.

Please also see our response to comment #13 about Figure 1 and the pattern of age-related change in power across the first year for confirmation that the age-related changes in summed power we see in the study (logarithmic, nonlinear age-related increases) are consistent with the age-related changes observed in the limited studies that have examined EEG power across the same age-range.

Xiao, R., Shida-Tokeshi, J., Vanderbilt, D.L., Smith, B. 2018. Electroencephalography power and coherence changes with age and motor skill development across the first half year of life. PlosONE.

Hashemi, M., Hutt, A., Hight, D., Sleight, J. 2017. Anesthetic action on the transmission delay between cortex and thalamus explains the beta-buzz observed under propofol anesthesia. PlosONE.

Kirov, R., Weiss, C., Siebner, H.R., Born, J., Marshall, L. 2009. Slow oscillation electrical brain stimulation during waking promotes EEG theta activity and memory encoding. PNAS.

From Kirov et al., PNAS 2009:

(frontal EEG power distributions with peak in beta power in restingstate setting)

Methods section edit (lines 285 – 297):

“For each of the 6 frequency bands typically used in infant EEG studies, the summed power across all frequencies within the band was calculated as the measure of total power in that frequency band (i.e. power density, or power/Hz was not used in these analyses. Here, summed power in a frequency band is equivalent to the area under the power density curve for that frequency band; Figure 1). Frequency bands were defined as follows: delta, 2 – 3.99 Hz; theta, 4 – 5.99 Hz; low alpha, 6 – 8.99 Hz; high alpha, 9 – 12.99 Hz; beta, 13 – 29.99 Hz; gamma, 30 – 50 Hz. For each EEG recording and each channel, the average summed power in each frequency band across all 2-second segments was calculated and normalized by a log base 10 transform. For each EEG recording, the summed power in each frequency band was then averaged across all channels of interest. (See Figure 2 for the mean EEG summed power in each frequency band at each timepoint for the frontal configuration used in the primary analyses).”

Example results section edit (lines 465 – 466):

“EEG power for a given frequency band in these models reflects the summed power across all frequencies within that frequency band (summed power).”

Conceptual Figure (Panel A):

Figure 1: Analysis schematic

Conceptual diagram illustrating how longitudinal EEG parameters were generated and analyzed. **A (for each participant):** at every age, the total, summed power in each canonical frequency band (delta, theta, low alpha, high alpha, beta, gamma) was calculated as the area under the curve of the EEG power distribution (left panel). Growth trajectories of summed power in each frequency band were generated across 1) 3-12, 2) 12-24, and 3) 3-36 postnatal months of age (beta frequency band 3 – 36 month trajectories plotted here, middle panel). Growth trajectories of summed power from each frequency band were linearized by modeling log (summed power) as a function of log (age) for each frequency band; this allowed for the calculation of an estimated intercept (here, at 6 months) and a linear developmental slope for each individual with at least 2 EEG recordings to be submitted to group-level analysis (right panel, here for beta frequency band).

Supplemental Figure 2 (reproduced in lower resolution here):

Power Spectra from Frontal Electrodes

Supplemental Figure 2: EEG Power Distributions Across Age

The distribution of log-transformed power for all groups, low-risk control (LRC), high-risk without ASD (HRA-), and high-risk with ASD (ASD), at each age (in months) for the frontal region of interest.

28. Separating the power trajectory plots by group in Figure 1 and presenting all the frequency bands together on one plot lessens the ability to distinguish between group differences by frequency band. As presented, the small differences in trajectory are difficult to decipher beyond the general impression that they all look quite similar across groups. It might be more beneficial to present a plot for each frequency band with different lines for each group in order to make more direct comparisons for intercept and trajectory (similar to the Tierney paper referenced above).

We agree that Figure 1 should be re-organized to help detect the differences picked up in the statistical analyses, and we have reformatted the figure as the reviewer suggests, so that each frequency band is plotted with different lines for each group (please see response to comment #13 for the revised figure).

REVIEWERS' COMMENTS:

Reviewer #1 (Remarks to the Author):

The authors did an excellent job responding to reviewer comments.

Reviewer #2 (Remarks to the Author):

The revised manuscript addressed my earlier concerns.

Minor edits:

The newly added second electrode cluster used in the supplemental analysis is inconsistently labeled as either "temporal-parietal" (the main text and Table 2) vs. "temporal" (in Supplemental Figure 1 and in Supplemental Tables 4-6).

In Supplemental Figure 1, "Sensory" should be "Sensor" in the name of the 64-channel net.

Line 275: "from the a priori frontal cortex region" - there appears to be a word missing after 'a priori' (e.g., selected/determined/etc.?)

Reviewer #3 (Remarks to the Author):

The authors have provided a thorough and thoughtful response to my comments and concerns. The clarifications on methodology have largely alleviated my concerns. While the 82% good channels threshold is more liberal than I would prefer, I do understand the difficulty of recording clean data in infants and this is not a hill I am prepared to die on (or let the paper die on) as the study overall is strong and the groups do not differ on this parameter.

We thank the Reviewer for their further feedback. Below, we have addressed each of the remaining suggestions for the revised manuscript.

Reviewer 2:

Minor edits:

The newly added second electrode cluster used in the supplemental analysis is inconsistently labeled as either "temporal-parietal" (the main text and Table 2) vs. "temporal" (in Supplemental Figure 1 and in Supplemental Tables 4-6).

We have edited the figure caption and tables to refer to the electrode cluster as "temporal-parietal" consistently throughout the submission.

Supplemental Figure 1 legend excerpt:

"... and the temporal-parietal (abbreviated to T-P in the figure) ROI shown in orange..."

In Supplemental Figure 1, "Sensory" should be "Sensor" in the name of the 64-channel net.

We have edited the figure legend to correct this mistake.

"The 128-channel EGI HydroCel Geodesic Sensor Net (version 1.0), top panel, and 64-channel EGI Geodesic Sensor Net (version 2.0), ..."

Line 275: "from the a priori frontal cortex region" - there appears to be a word missing after 'a priori' (e.g., selected/determined/etc.?)

We have edited the text to correct this omission.

"Supplemental analyses were performed to compare the results from the a priori selected frontal cortex region of interest with both..."